# Chromatin accessibility promotes hematopoietic and leukemia stem cell activity

Lucia Cabal-Hierro[1,2], Peter van Galen [2,3], Miguel A. Prado [4], Kelly J. Higby[1,2], Katsuhiro Togami[1,2], Cody T. Mowery [1,2], Joao A. Paulo [4], Yingtian Xie[1,5], Paloma Cejas[1,5], Takashi Furusawa[6], Michael Bustin [6], Henry W. Long [1,5], David B. Sykes[7], Steven P. Gygi[4], Daniel Finley[4], Bradley E. Bernstein [2,3] & Andrew A. Lane [1,2✉]

Chromatin organization is a highly orchestrated process that influences gene expression, in part by modulating access of regulatory factors to DNA and nucleosomes. Here, we report that the chromatin accessibility regulator HMGN1, a target of recurrent DNA copy gains in leukemia, controls myeloid differentiation. HMGN1 amplification is associated with increased accessibility, expression, and histone H3K27 acetylation of loci important for hematopoietic stem cells (HSCs) and leukemia, such as HoxA cluster genes. In vivo, HMGN1 overexpression is linked to decreased quiescence and increased HSC activity in bone marrow transplantation. HMGN1 overexpression also cooperates with the AML-ETO9a fusion oncoprotein to impair myeloid differentiation and enhance leukemia stem cell (LSC) activity. Inhibition of histone acetyltransferases CBP/p300 relieves the HMGN1-associated differentiation block. These data nominate factors that modulate chromatin accessibility as regulators of HSCs and LSCs, and suggest that targeting HMGN1 or its downstream effects on histone acetylation could be therapeutically active in AML.

[1] Department of Medical Oncology, Dana-Farber Cancer Institute, Harvard Medical School, Boston, MA, USA. [2] Broad Institute of Harvard and MIT, Cambridge, MA, USA. [3] Department of Pathology, Massachusetts General Hospital, Harvard Medical School, Boston, MA, USA. [4] Department of Cell Biology, Harvard Medical School, Boston, MA, USA. [5] Center for Functional Cancer Epigenetics, Dana-Farber Cancer Institute, Boston, MA, USA. [6] Laboratory of Metabolism, National Cancer Institute, Bethesda, MD, USA. [7] Center for Regenerative Medicine, Massachusetts General Hospital, Boston, MA, USA. ✉email: andrew_lane@dfci.harvard.edu

In eukaryotic cells, DNA is packaged into chromatin, a dynamic structure containing genomic DNA, histones, and nonhistone proteins. Chromatin regulation by chemical modification of DNA or histones modulates gene expression and is important for development, differentiation, and response to signals[1]. Cancer cells frequently acquire abnormalities in the same chromatin regulatory pathways, which promote malignancy and may impart targetable therapeutic dependencies[2]. Relatively less is known about how nonhistone nucleosome-associated proteins that modify chromatin compaction and accessibility contribute to transformation.

Chromatin accessibility affects gene expression by altering interactions of transcriptional regulatory factors with their targets. The linker histone H1 organizes nucleosomes into higher-order structures, increasing chromatin folding, and is associated with decreased transcription[3,4]. In contrast, the HMGN family of nucleosome binding proteins competes with histone H1 and promotes decompaction of chromatin, increased accessibility, and locally enhanced transcription[5,6]. HMGNs interact with nucleosomes independent of DNA sequence, via a 20 amino acid nucleosome binding domain that recognizes an acidic patch formed by the core histones H2A and H2B[7]. HMGN genomic positioning is not random; it preferentially co-localizes with regulatory marks at active promoters and affects nucleosome organization, DNase hypersensitivity patterns, and post-translational histone marks[8,9]. Despite these links between HMGNs and chromatin regulation, a mechanistic understanding of how HMGNs contribute to disease is not as well understood.

Hematopoiesis is a highly regulated process requiring both self-renewal and differentiation of hematopoietic stem cells (HSCs) to maintain diverse mature blood components. Disruption of normal hematopoiesis can lead to pathologic states, including leukemia[10]. One hallmark of leukemia is a block in normal differentiation from hematopoietic stem/progenitor cells (HSPCs) to their mature progeny. In acute promyelocytic leukemia (APL), "differentiation therapy" using all-trans retinoic acid and arsenic trioxide releases the differentiation block and is curative in most patients, but treatments to promote differentiation have not emerged clinically in most other leukemia subtypes[11].

Amplification of chromosome 21, particularly the distal segment 21q22, is highly associated with acute leukemia. Germline amplification, such as in Down syndrome (trisomy 21), confers a markedly increased risk of myeloid and lymphoid leukemia[12]. In addition, somatic gain of chromosome 21 is associated with aggressive leukemias, such as iAMP21 B cell acute lymphoblastic leukemia that has multiple extra copies of 21q22 and a very poor prognosis[13]. In acute myeloid leukemia (AML) with complex karyotype, a subtype associated with poor outcomes, 21q22 is the most frequent region of high-level DNA copy gain[14]. We previously performed an RNA interference screen in murine B cell progenitors modeling Down syndrome and found that Hmgn1 was the amplified gene most critical to support hematopoietic colony forming activity[15]. In B cells, HMGN1 overexpression promotes global changes in transcription with selective amplification of lineage-specific survival pathways[16]. However, how 21q22/HMGN1 amplification affects HSPCs/myeloid differentiation or confers therapeutic vulnerability is not clear.

Here, we find that HMGN1 impairs normal myeloid differentiation in association with increased gene expression and H3K27 acetylation, particularly at promoters of genes that regulate HSPC identity and function. Moreover, HMGN1 overexpression promotes a clonal advantage in HSPCs in vivo and increases leukemia stem cell (LSC) activity in concert with AML oncogenes. Suggesting potential therapeutic relevance, the differentiation impairment by HMGN1 is dependent on the histone acetyltransferases (HATs) CBP and p300 and is reversible by HAT inhibition.

## Results

**HMGN1 overexpression impairs myeloid differentiation.** *HMGN1* is highly expressed across human and mouse immature hematopoietic stem and progenitor populations but is markedly downregulated in differentiated myeloid cells such as neutrophils and monocytes (Supplementary Fig. 1a)[17]. This is consistent with data from other tissues where downregulation of *HMGN1* is linked with differentiation to specific lineages[18]. Furthermore, when examined microscopically, hematopoietic progenitors, and AML blasts have visibly "open" chromatin, which compacts during normal myeloid development or after AML treatments that restore myeloid differentiation[19,20]. This led us to hypothesize that HMGN1's role in maintaining open chromatin might contribute to the differentiation block in AML.

To interrogate the role of 21q22 amplification and HMGN1 in myeloid differentiation, we immortalized primary hematopoietic progenitors in an ex vivo culture system that facilitates analysis of immature myeloid cells and their progeny during synchronized differentiation[21]. In mice, *Hmgn1* is located on chromosome 16 and is trisomic in several models of Down syndrome, including Ts1Rhr[22], which triplicates 31 genes orthologous to a segment of human chr21q22 that is recurrently amplified in AML. We transduced bone marrow from wild-type (WT), Ts1Rhr, and HMGN1-OE mice (a transgenic model only overexpressing human HMGN1, at 2–3 times the level of the endogenous protein[15,16,23]) with a retrovirus expressing an estrogen receptor (ER)-HoxB8 fusion protein, which maintains cells as immature progenitors in the presence of estradiol (E2). Upon removal of E2 and in the presence of interleukin 3, wild-type cells undergo synchronized differentiation to mature myeloid cells (CD11b+ GR-1+) over 6–7 days. In contrast, cells from the Ts1Rhr or HMGN1-OE models had delayed myeloid differentiation, as measured by later acquisition of CD11b and GR-1 (Fig. 1a, upper panel). Ts1Rhr and HMGN1-OE progenitors did not acquire mature myeloid cell morphology at day 4 (Fig. 1b) nor did HMGN1-OE progenitors upregulate reactive oxygen species (ROS) production during differentiation to the same degree as wild-type cells (Fig. 1c). This suggests that the HMGN1-associated differentiation abnormality was functionally relevant and not simply a change in cell surface marker expression. Ts1Rhr and HMGN1-OE undifferentiated progenitors also had an increased growth rate in the presence of E2 compared to wild-type cells (Fig. 1a, lower panel).

To validate the role of HMGN1 in myeloid differentiation, we performed knock-down and overexpression experiments in these model systems. First, we found that HMGN1 was necessary for the aberrant differentiation phenotype in the 21q22 triplication model, because shRNA knockdown of *Hmgn1* in Ts1Rhr cells promoted increased differentiation (Fig. 1d). Next, we transduced WT progenitors with retroviral constructs to express either wild-type HMGN1 (HMGN1-WT) or a mutant unable to bind to nucleosomes (HMGN1-SEmut)[16]. Overexpression of HMGN1, but not the HMGN1-SE mutant, caused increased proliferation and impaired differentiation after removal of E2 (Fig. 1e).

We next asked if modulating HMGN1 was consequential in human AMLs. Among 216 human cancer cells lines analyzed in Project Achilles v2.4.3, a genome-wide RNA interference screen[24], there were 18 derived from patients with AML. shRNAs targeting *HMGN1* were selectively depleted in AML cells compared to all other cancer cell types, suggesting a lineage-specific dependency on *HMGN1* in AML (Supplementary Fig. 1b). AML cell lines trended toward higher expression of *HMGN1* compared to other lineages (Supplementary Fig. 1c). However, it is important to note that most of the AML cell lines tested were diploid for 21q22/*HMGN1* and none have known mutations in

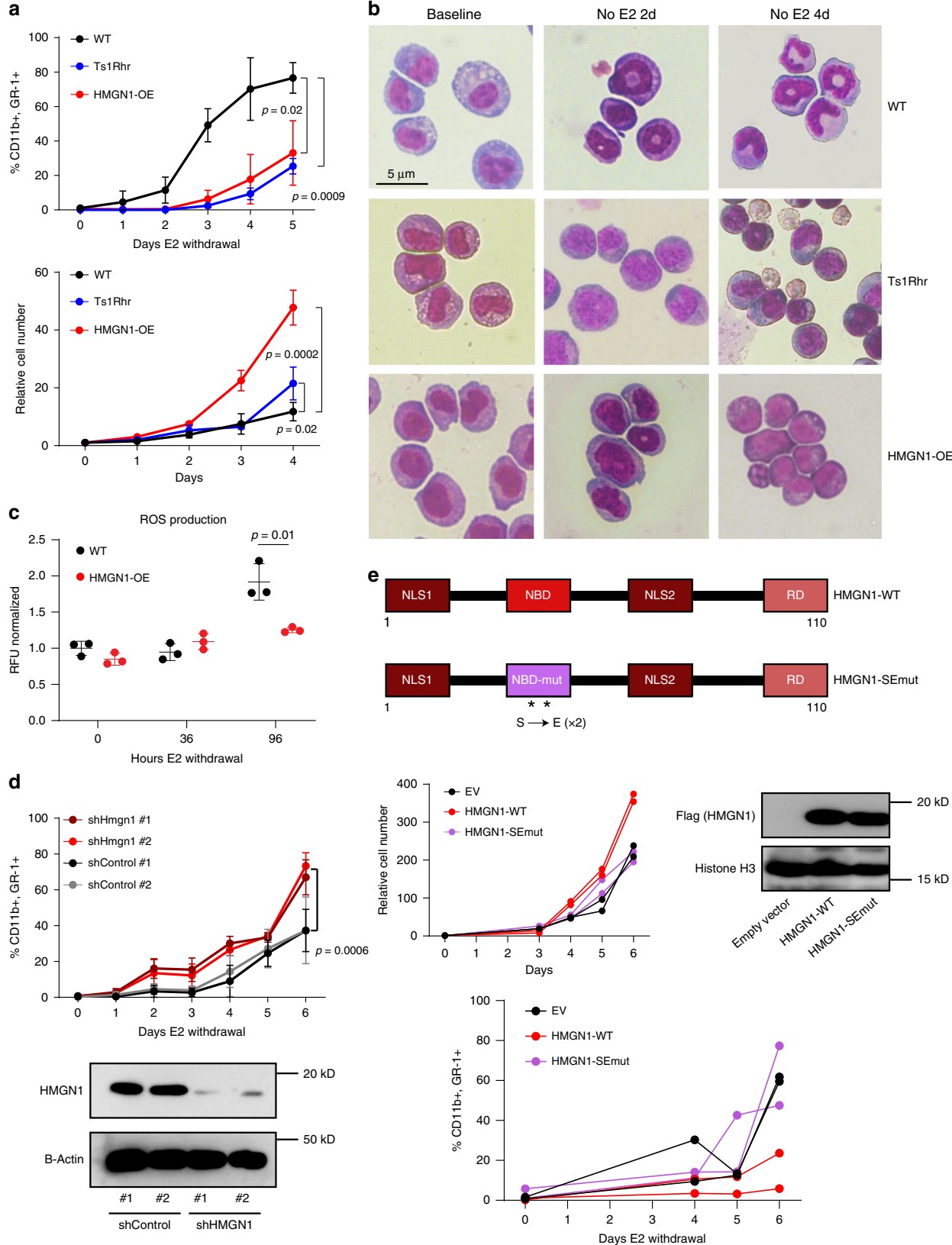

*HMGN1*, yet they still exhibited relative *HMGN1* dependency (Supplementary Fig. 1d), raising the possibility that HMGN1 is important for AML even in cases without chr21 amplification. To confirm these findings by an orthogonal technique, we used CRISPR/Cas9 to deplete HMGN1 from AML cell lines. Consistent with the Project Achilles data, loss of HMGN1 was

associated with decreased proliferation and/or induction of differentiation (Supplementary Fig. 2).

**HMGN1 promotes expression of stem cell and leukemia genes.**
Our next goal was to determine how HMGN1 overexpression

**Fig. 1 HMGN1 overexpression impairs myeloid differentiation. a** Analysis of myeloid differentiation (upper) and proliferation (lower) in wild type, Ts1Rhr, and HMGN1-OE myeloid progenitors. Differentiation was measured after withdrawal of E2 as percentage of cells expressing CD11b and GR-1. **b** Representative morphology of progenitors at baseline, 2, and 4 days after withdrawal of E2. **c** Analysis of reactive oxygen species (ROS) production during differentiation. Plot is represented as relative fluoresce units (RFU) normalized to wild type at time 0. **d** Myeloid differentiation with or without shRNA knockdown of *Hmgn1* in Ts1Rhr cells, Data compared by two-sided *t* test between control and *Hmgn1* hairpins. Western blot for HMGN1 96 h after induction of the indicated shRNAs. **e** (Upper) Schematic representation of HMGN1-WT and HMGN1-SEmut (nucleosome binding domain (NBD) mutant harboring S20E and S24E mutations). NLS nuclear localization signal, RD relaxation domain. Proliferation (middle) and differentiation (lower) assays were measured as in panel a) in wild-type cells expressing the indicated proteins (EV empty vector). Western blot reflects expression of FLAG-HMGN1-WT and FLAG-HMGN1-SEmut. In all panels, graphs represent *n* = 3 biologically independent samples, except panel **e**, which is biologically independent duplicates. Conditions compared by two-sided *t* test. Data are presented as mean values ± SD. Source data are provided as a Source Data file.

affected the transcriptome and proteome of myeloid cells. We first determined the "point of no return" after removal of E2 beyond which wild-type cells were fully committed to terminal myeloid differentiation. Thirty-six hours after removal of E2 was the first timepoint where differentiation could no longer be reversed by re-addition of E2 (Supplementary Fig. 3). By 96 h after removal of E2 there were obvious differences in proliferation between wild-type and HMGN1-OE cells (Fig. 1a). Therefore, we performed RNA-sequencing and proteome analysis via tandem mass tagging (TMT) at 0, 36, and 96 h after the withdrawal of E2 from wild-type and HMGN1-OE cultures. Transcriptome analysis was normalized to "spiked-in" synthetic RNAs to control for possible differences in total RNA content per cell[25]. We also performed an Assay for Transposase-Accessible Chromatin using sequencing (ATAC-seq) to measure chromatin openness, and multiplexed, indexed T7 chromatin immunoprecipitation sequencing (Mint-ChIP)[26] to profile histone modifications associated with HMGN1 overexpression (Fig. 2a).

First, we confirmed that differential RNA expression was correlated with proteomic changes at all three time points of differentiation, suggesting that the HMGN1-associated transcriptional changes were also reflected in protein abundance (Supplementary Fig. 4a, b). Gene set enrichment analysis (GSEA) of RNA expression in immature progenitors (in E2) suggested that myeloid differentiation programs were reduced in HMGN1-OE cells compared to wild-type cells (Fig. 2b, Supplementary Data 1). Furthermore, HMGN1-OE progenitors were more transcriptionally similar to HSCs or AML LSCs[27] when compared to wild-type progenitors (Fig. 2c, Supplementary Data 2). This transcriptional difference is notable given that the cells are morphologically similar (Fig. 1b) and that wild-type, Ts1Rhr, and HMGN1-OE undifferentiated cells have a comparable cell surface phenotype, most similar to that of a normal granulocyte–monocyte progenitor (GMP, Lineage-CD117+Sca1-CD34+CD16/32+, Supplementary Fig. 5a). Among the most upregulated genes in HMGN1-OE cells were the HoxA and HoxB families. This difference was present in the undifferentiated state and persisted during differentiation (Fig. 2c, d, Supplementary Data 3). Hox proteins are homeobox transcription factors that are well-known regulators of hematopoietic stem and progenitor cell (HSPC) function[28] and have been implicated as oncogenes in AML[29]. We validated increased expression of *HoxA7* and *HoxA9* in HMGN1-OE cells by Q-RT-PCR (Supplementary Fig. 5b).

ATAC-seq revealed that HMGN1 overexpressing myeloid progenitors had globally increased chromatin accessibility compared to wild-type cells (Fig. 2e, Supplementary Fig. 5c). This increased ATAC-seq signal was particularly evident in the HoxA and HoxB clusters that were identified as targets of higher RNA expression in HMGN1-overexpressing cells (Fig. 2f). We hypothesized that increased chromatin accessibility might also lead to changes in histone marks, consistent with what has been observed with HMGN1 expression in fibroblasts[30]. Therefore, we profiled chromatin marks during myeloid differentiation in the

context of HMGN1 overexpression using Mint-ChIP. Mint-ChIP enables quantitative measurements of histone modifications by normalization to total histone H3[26]. Overexpression of HMGN1 was associated with global increased levels of H3K27 acetylation (H3K27ac), a marker of active transcription, at transcription start sites (TSSs) compared to wild-type cells, and this increase was maintained during induction of myeloid differentiation (Fig. 2g). Higher global H3K27 acetylation was validated by western blotting and flow cytometry across multiple time points during differentiation (Fig. 2g).

Although there was a global increase in chromatin accessibility and H3K27ac across the genome in undifferentiated cells, not all loci were affected equally, suggesting the possibility of additional specificity to the HMGN1 effect. Therefore, we performed an integrated analysis of RNA-seq, ATAC-seq, and Mint-ChIP, focusing on genes at the intersection of the most differentially expressed with the largest changes in accessibility and histone marks at their promoters. *HoxA3*, *HoxA7*, and *HoxA9* ranked as those with the most significant overlapping differences between wild-type and HMGN1-OE cells (Fig. 3a, b, Supplementary Fig. 5d). Loss of the repressive histone mark H3K27 trimethyl (H3K27me3), which is an expected reciprocal change in the setting of increased H3K27 acetylation, was also detected in the HoxA cluster (Fig. 3a). We also observed dose-dependent binding of HMGN1 to chromatin at the *HoxA7* and *HoxA9* genes, in both HMGN1-OE compared to wild-type cells and in progenitors retrovirally overexpressing wild-type HMGN1 compared to a nucleosome-binding incompetent mutant (Fig. 3c). Notably, nearly identical positions in the orthologous human HoxA cluster were previously implicated as a region of concentrated epigenetic regulatory activity in AML[31].

HMGN1 is known to facilitate HAT activity in local chromatin environments by increasing accessibility to chromatin-modifying enzymes[30]. We found that many targets of the HATs CBP and p300 in murine hematopoietic cells[32] were upregulated in HMGN1-OE progenitors, including several genes important in HSCs and leukemia (e.g., *HoxA and Hox B* family, *Meis1*, and *Msi2*) (Fig. 3d; left, CBP targets; right, p300 targets). We confirmed these associations using ChIP-PCR, which validated the Mint-ChIP findings of increased H3K27ac and decreased H3K27me3 at the *HoxA7* and *HoxA9* loci and also demonstrated enriched binding of CBP in the *HoxA7* locus and CBP and p300 in the *HoxA9* locus in HMGN1-OE cells (Fig. 3e). We also confirmed that HMGN1 levels correlated with H3K27ac, and CBP and p300 binding in *HoxA7* and *HoxA9* loci in Ts1Rhr myeloid progenitors with and without knockdown of *Hmgn1* (Fig. 3f, Supplementary Fig. 5e). Together, these data suggest that the effects of HMGN1 in myeloid progenitors are most prominent at genes that are important in myeloid differentiation, HSC function, and leukemia, including but not limited to, Hox transcription factors.

**HMGN1 promotes clonal expansion and HSC activity in vivo.** Given its striking effects on differentiation and stem cell

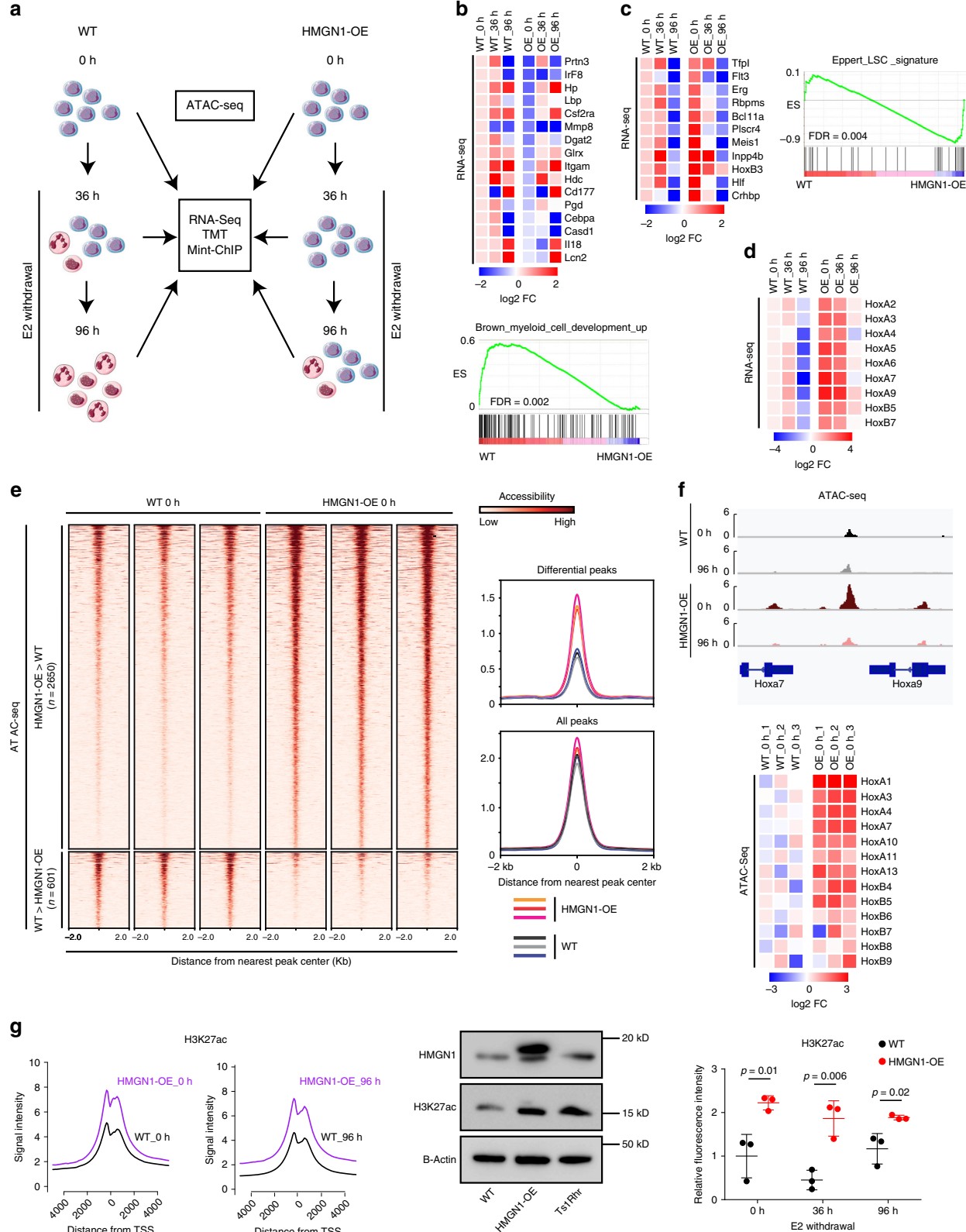

programs, we next asked how HMGN1 overexpression affects hematopoiesis and HSPC function in vivo. First, we analyzed HSPCs in HMGN1-OE and wild-type mice at 3 and 12 months of age. No significant differences were observed in young mice. However, older HMGN1-OE animals had higher frequency of immature, lineage-negative (Lin−) cells, Lin-CD117/c-Kit+ (LK) myeloid progenitors, and expansion of GMPs (Fig. 4a).

To further explore the role of HMGN1 in hematopoiesis, we performed serial competitive bone marrow transplantation between HMGN1-OE (marked by CD45.2) and CD45.1(STEM) wild-type marrow (Fig. 4b). CD45.1(STEM) is a C57BL/6 congenic strain with a single point mutation that confers recognition by CD45.1-specific monoclonal antibodies and that lacks the competitive disadvantage inherent in the B6.SJL-Ptprc

**Fig. 2 HMGN1 promotes chromatin accessibility and confers stem cell and leukemia-associated transcriptional and chromatin phenotypes. a** Schematic of the experimental approach for ATAC-seq, RNA-seq, Mint-ChIP chromatin profiling, and TMT proteomics during myeloid differentiation. **b** Gene set enrichment analysis (GSEA) of RNA-seq in undifferentiated wild-type and HMGN1-OE progenitors showing enrichment of the Brown_myeloid_cell_ development_up gene set in wild-type cells. Heatmap shows mean expression of the leading edge genes from the gene set in GSEA in biological duplicates of wild-type and HMGN1-OE cells at the indicated time points, expressed as log2 fold change (FC) relative to WT_0h. **c** GSEA of wild-type and HMGN1-OE progenitors showing enrichment of genes in the Eppert_LSC_signature gene set in HMGN1-OE cells. Heatmap of leading edge genes as in panel **b**. **d** Heatmap of expression of Hox cluster genes in wild-type and HMGN1-OE cells during differentiation, expressed as in panel **b**. **e** Chromatin accessibility tracks representing distance from nearest peak center of all differential peaks in ATAC-seq (left), and metagene plots of ATAC-seq differential peaks and all shared peaks in wild-type and HMGN1-OE cells (right, $n = 3$ biologically independent replicates). **f** Gene tracks showing ATAC-seq reads at the *HoxA7* and *HoxA9* loci at baseline and 96 h in wild-type and HMGN1-OE progenitors (top). Heatmap of ATAC-seq signals at HoxA and HoxB genes in wild-type and HMGN1-OE progenitors, expressed as log2 FC relative to the mean WT_0h value (bottom). **g** Metagene profiles of H3K27 acetylation surrounding promoters in wild-type and HMGN1-OE cells at 0 and 96 h of differentiation (left). H3K27ac levels were also measured by western blotting (middle) and by intracellular flow cytometry (right, normalized to WT 0 h, $n = 3$ biological replicates, genotypes compared by two-sided $t$ test). Data are presented as mean values ± SD. Source data are provided as a Source Data file.

(a)Pepc(b)/BoyJ-CD45.1 strain[33]. HMGN1-OE (CD45.2) transplanted cells had an increasing representation among mature blood cells over three serial bone marrow transplantations, indicating a progressive HSPC competitive advantage (Fig. 4c). Four months after the initial transplantation, mice were sacrificed for analysis of CD45.2 (HMGN1-OE) and CD45.1 (wild type) representation in bone marrow populations (Fig. 4b). In the primary transplant, we did not see a significant difference in the overall mature peripheral blood cell CD45.2/CD45.1 ratio after 4 months (Fig. 4c, first Tx). However, in bone marrow at the same time point, an advantage was present for HMGN1-OE hematopoietic progenitors, including in early multipotent progenitors (MPPs) (Fig. 4d). In secondary and tertiary transplants (second Tx and third Tx), the HMGN1-OE HSPC competitive advantage progressively increased in all primitive and mature hematopoietic populations, including in the peripheral blood, suggesting a long-term HSC advantage associated with HMGN1 overexpression (Fig. 4c, d).

To understand mechanisms underlying the HMGN1-OE competitive advantage and to determine if the epigenome changes we observed during ex vivo myeloid differentiation were also seen in intact primary hematopoiesis, we analyzed transcriptomes and histone marks from sorted HMGN1-OE and wild-type HSPCs that were in competition in vivo. GSEA of differentially expressed genes in LK cells revealed gene sets associated with mitochondrial activity enriched in HMGN1-OE cells (Supplementary Fig. 6a, b, Supplementary Data 4). Mitochondrial biogenesis is triggered with cell cycle induction in HSCs exiting quiescence and is linked to proliferation[34,35]. We also observed decreased expression of programs associated with HSC quiescence[36] and increased expression of genes upregulated in cycling HSCs[37] in HMGN1-OE HSPCs (Supplementary Fig. 6c, d, Supplementary Datas 5 and 6). Therefore, we tested whether HMGN1 overexpression was associated with decreased quiescence in HSPCs in vivo. We found that G0 quiescent subpopulations were reduced in both LSK and LK stem/progenitor cells from HMGN1-overexpressing animals (Fig. 4e). These data indicate that HMGN1 overexpression may provide a competitive advantage to HSPCs associated with mitochondrial biogenesis and proliferation.

In most cases where alterations in cell cycle regulators provide a short-term HSPC proliferative advantage, the long-term consequence is premature exhaustion of HSC self-renewal in vivo[38]. However, HMGN1-OE HSPCs had both reduced quiescence and increased repopulating activity in serial transplantation, suggesting they had sustained self-renewal. In support of that hypothesis, HMGN1-overexpressing HSPCs had increased expression of markers of HSC activity, such as *HoxA7* and *HoxA9*, and simultaneously decreased expression of genes associated with myeloid differentiation (Supplementary Fig. 6e,

Supplementary Data 7). Expression differences were also associated with changes in H3K27ac at TSSs of genes identified by GSEA (Supplementary Fig. 6f). Together, the gene expression and epigenomic changes in HMGN1-OE bone marrow HSPCs are similar to those we observed in progenitors studied by ex vivo myeloid differentiation (Fig. 2), supporting the in vivo relevance of HMGN1 effects. Similarly, we found that c-Kit+ myeloid progenitors from HMGN1-OE bone marrow had delayed differentiation compared to wild-type cells when cultured in vitro (Fig. 4f). Collectively, these data support the role of HMGN1 in promoting clonal expansion in vivo associated with expression changes in genes involved in stemness, quiescence, and differentiation programs.

**HMGN1 cooperates with AML oncogenes to increase LSC activity.** HMGN1 overexpression impaired myeloid differentiation and conferred a competitive advantage to HSPCs but did not cause leukemia on its own (up to 18 months of observation). Therefore, we asked if HMGN1 overexpression cooperated with AML oncogenes in our established assays of myeloid differentiation. We first measured differences in HMGN1-OE or wild-type myeloid progenitor growth and differentiation in vitro in the presence of several AML oncogenes. Some (BCR-ABL and FLT3-ITD), were unaffected by HMGN1 overexpression (Supplementary Fig. 7a). Others, such as AML-ETO9a, MLL-AF9, PML-RARα, NRAS G12V, GATA2 R396W, NPM1mut (exon 12 insertion[39]), and MOZ-TIF2 cooperated with HMGN1 as seen by enhanced myeloid differentiation delay compared to either alteration alone (Fig. 5a, Supplementary Fig. 7a). In addition, AML-ETO9a, NPM1mut, and MOZ-TIF2 cooperated with HMGN1 overexpression to increase colony numbers, suggesting enhanced self-renewal capacity. Increased colony formation by AML-ETO9a plus HMGN1-OE cells was maintained after multiple replating events (Fig. 5b, Supplementary Fig. 7b). This enhanced self-renewal was associated with a lower frequency of differentiated cells in HMGN1-OE plus AML-ETO9a colonies (Supplementary Fig. 7b).

We confirmed these findings in two additional independent primary cell models. First, we found that loss of one copy of *Hmgn1* in Ts1Rhr progenitors (Ts1Rhr_HMGN1+/−; three copies of the 21q22 genes, except only two copies of *Hmgn1*[16]) restored myeloid differentiation and that the differentiation block by AML-ETO9a was greater in Ts1Rhr compared to Ts1Rhr_HMGN1+/− cells (Fig. 5c). Second, in colony assays, depletion of *Hmgn1* in Ts1Rhr cells by deletion of one copy (Ts1Rhr_HMGN1+/−) or by shRNA decreased the self-renewal activity associated with AML-ETO9a (Supplementary Fig. 8). Cooperation between HMGN1 and these specific fusion oncogenes is interesting given that AML-ETO promotes

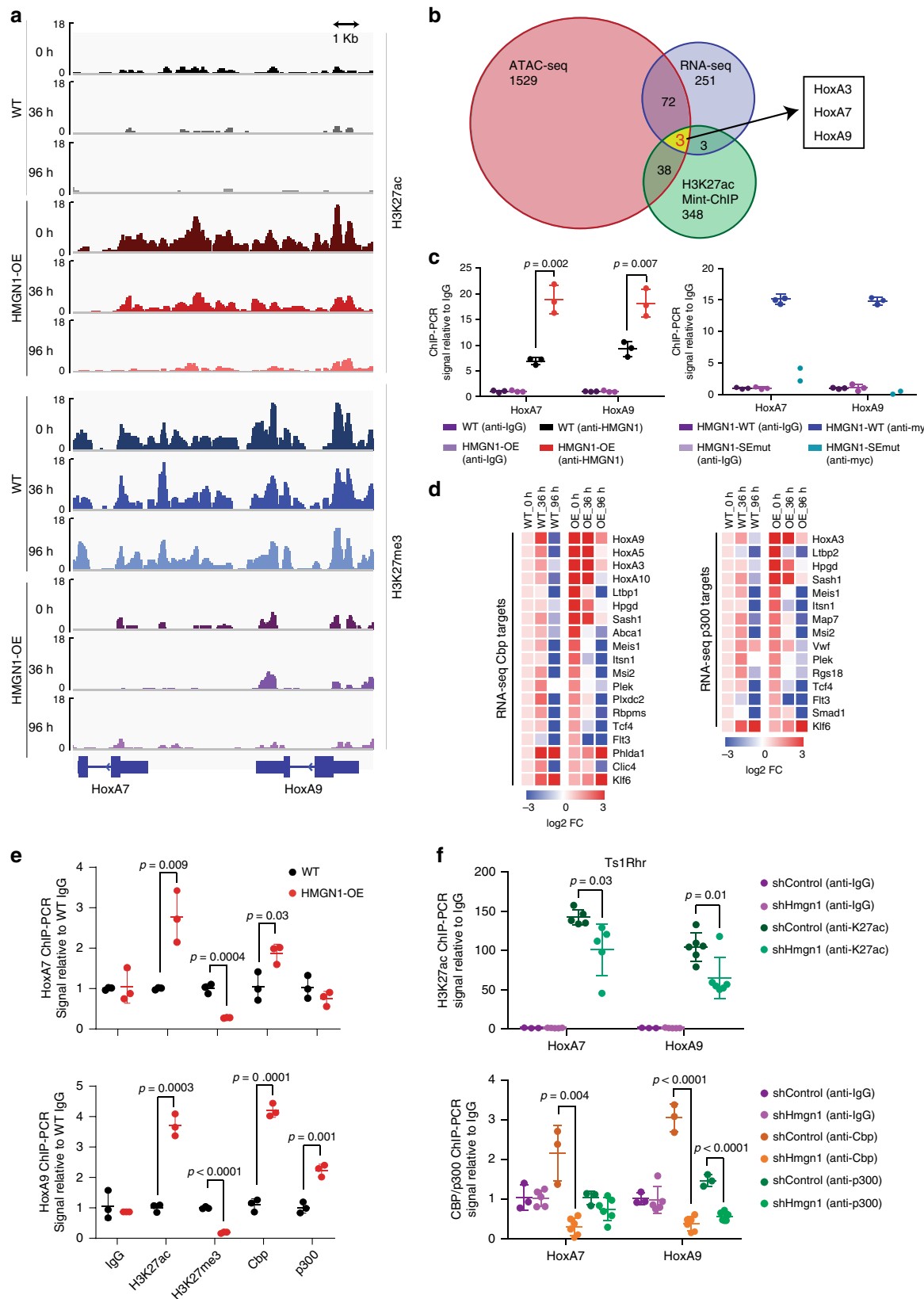

HSPC self-renewal and leukemia via CBP/p300 acetylation[40] and MOZ is a HAT that depends on HAT activity to inhibit senescence[41]. For these reasons, we hypothesized that AML-ETO9a and HMGN1 overexpression would cooperate in vivo.

We enriched CD117+ myeloid progenitors from 8-week-old wild-type or HMGN1-OE bone marrow and transduced the cells with a retrovirus expressing AML-ETO9a linked to GFP or with a GFP-only control, and then transplanted equivalent numbers of transduced cells into irradiated recipients (Fig. 5d). Two months

**Fig. 3 Chromatin and expression changes of Hox genes linked to HMGN1 are associated with the histone acetyl transferases (HATs) Cbp and p300.**
**a** Gene tracks showing H3K27ac and H3K27me3 Mint-ChIP reads at the *HoxA7* and *HoxA9* loci at baseline and during differentiation in wild-type and HMGN1-OE progenitors. **b** Venn diagram overlapping RNA-seq, H3K27ac Mint-ChIP, and ATAC-seq sets of significant differences in HMGN1-OE vs. wild-type progenitors at baseline (log2FC > 1.2, $p < 0.05$). **c** ChIP-PCR in wild-type and HMGN1-OE myeloid progenitors (left) and wild-type progenitors overexpressing either myc-tagged HMGN1-WT or HMGN1-SEmut (right) for relative HMGN1 binding in the *HoxA7* and *HoxA9* loci. $n = 2$ or 3 biologically independent samples, as indicated. **d** Heatmap of expression of CBP (left) and p300 (right) target genes that are members of the Wong_Adult_Tissue_ Stem_Cell gene set and enriched in HMGN1-OE progenitors, expressed as log2 fold change (FC) relative to WT_0 h. **e** ChIP-PCR in wild-type and HMGN1-OE myeloid progenitors for relative H3K27ac, H3K27me3, Cbp, and p300 binding in the *HoxA7* and *HoxA9* loci. $n = 3$ biologically independent samples. **f** ChIP-PCR in Ts1Rhr progenitors with *Hmgn1* knockdown by shRNA for relative H3K27ac (top) and the HATs Cbp and p300 (bottom) in the *HoxA7* and *HoxA9* loci. $n = 3$ biologically independent samples, each measured once (conditions with three data points) or twice (six data points). All statistical comparisons are by two-sided *t* test. Data are presented as mean values ± SD. Source data are provided as a Source Data file.

after injection, we sacrificed a subset of animals from each group and analyzed the relative frequency of hematopoietic progenitor and mature populations. We did not observe any significant differences between recipients of AML-ETO9a-transduced wild-type and HMGN1-OE bone marrow (Supplementary Fig. 9a). However, upon culturing CD117+ cells under conditions that promote myeloid differentiation, we observed moderately increased expansion of stem/progenitor populations and decreased production of CD11b+ mature myeloid cells in HMGN1-OE cultures (Supplementary Fig. 9b).

Beginning at approximately 100 days after transplantation, recipients of AML-ETO9a-transduced marrow began to display signs of systemic illness. Moribund animals had elevated frequency of GFP+ cells in the peripheral blood, bone marrow, and in the spleen, accompanied by splenomegaly (Supplementary Fig. 9c). Although the overall survival was not significantly different in the primary transplant recipients of HMGN1-OE plus AML-ETO9a compared to wild-type plus AML-ETO9a (Fig. 5e), we tested for phenotypic differences in the diseases. HMGN1-OE leukemias were enriched for LSK-like cells and particularly for the LSK subset having a multipotent progenitor (MPP)-like phenotype (Lin− CD117+ Sca-1+ CD150+ CD48−, Fig. 5f). This suggested that HMGN1-overexpressing leukemias had a larger proportion of primitive, stem-like cells compared to wild-type leukemias. Moreover, similar to what we had observed in HMGN1-OE ex vivo immortalized progenitors and HSPCs in vivo, HMGN1-OE plus AML-ETO9a leukemias had increased H3K27 acetylation, particularly within the subset of leukemia cells that had an HSC-like phenotype (Fig. 5g).

The expansion of phenotypically immature leukemia cells led us to hypothesize that HMGN1 overexpressing leukemias might be associated with increased LSC activity. Consistent with this prediction, in long-term culture initiating cell (LTC-IC) assays, an in vitro measure of HSPC activity that also correlates with survival in human AML[42], HMGN1-OE plus AML-ETO9a leukemias had increased colony formation capacity (Fig. 5h). To measure LSC activity in vivo, we performed limiting dilution transplantation of leukemias into secondary recipients. When we examined the peripheral blood 4 weeks after transplantation, while all animals had a large AML-ETO9a/GFP+ transplanted cell fraction (mean ~80%), we observed an expanded lineage-negative primitive hematopoietic population averaging 59% of the GFP+ mononuclear cells selectively in HMGN1-OE plus AML-ETO9a recipients (Fig. 5i). In contrast, recipients of equivalent cell doses of wild-type plus AML-ETO9a secondary transplants did not have evidence of this circulating undifferentiated population and nearly all the GFP+ cells had a CD11b+ GR-1+ mature myeloid surface phenotype (Fig. 5i). HMGN1-OE plus AML-ETO9a transplants caused fatal leukemia within 75 days in 8/8 secondary recipients that received 100,000 GFP+ cells, 8/8 that received 10,000 cells,

and 6/8 that received 1000 cells, which correlates with an estimated leukemia-initiating cell frequency of 1:721 (95% CI 1:304–1:1714) (Fig. 5j). At the same time point, no recipients of wild-type plus AML-ETO9a splenocytes at the same doses developed fatal leukemia. Thus, HMGN1 overexpression increased LSC frequency and capacity to transplant a fatal HSPC-like immature leukemia in AML-ETO9a cells.

**Inhibition of HATs antagonizes HMGN1 effects**. We next explored therapeutic implications of HMGN1-associated myeloid phenotypes. Given the striking increase in H3K27ac at HAT binding sites within loci of genes important for HSPCs and leukemia (Fig. 2), we evaluated the effect of genetic or chemical HAT inhibition in primary cell models. We hypothesized that reversing H3K27 hyperacetylation might counteract the HMGN1-associated differentiation block. First, we used a CRISPR-Cas9 approach to target *Cbp* or *Ep300* in wild-type and HMGN1-OE myeloid progenitors and confirmed decreased expression (Fig. 6a). Upon removal of E2, whereas neither *Cbp* nor *Ep300*-targeted sgRNAs affected wild-type cell differentiation, disruption of either gene promoted differentiation in HMGN1-OE cells (Fig. 6b). Targeting of *Cbp* or *Ep300* also decreased the aberrant global hyperacetylation of H3K27 in HMGN1-OE progenitors (Fig. 6c). The pro-differentiating effect of HAT targeting in HMGN1-overexpressing cells was corroborated by morphologic assessment (Fig. 6d).

To determine whether pharmacologic targeting of acetyltransferases could also promote myeloid differentiation, we assessed the effects of small molecule inhibitors of HATs. Treatment with C646, an acetyl-CoA competitive inhibitor of CBP/p300[43], promoted myeloid differentiation in HMGN1-OE progenitors, without affecting wild-type cell differentiation (Fig. 6e). The more potent and structurally distinct catalytic HAT inhibitor A-485, but not its inactive analog A-486[44], also restored normal myeloid differentiation in HMGN1-OE progenitors, again without affecting wild-type cells (Fig. 6e). C646 and A-485-associated differentiation in HMGN1-OE cells was confirmed by morphologic assessment (Fig. 6f). C646 also decreased *HoxA7* and *HoxA9* expression, in association with decreased binding of CBP and p300 and reduced H3K27 acetylation at the *HoxA7* and *HoxA9* loci (Fig. 6g, Supplementary Fig. 10). In contrast, other small molecules targeting H3K27 mark placement or recognition had no genotype-selective effect on HMGN1-OE progenitor differentiation. These included inhibitors of the H3K27ac "reader" BET bromodomain proteins (JQ1[45], Cbp30[46], and iCbp112[47]); ATP citrate lyase (ACLY), the enzyme responsible for producing acetyl-CoA from citrate (BMS3031411[48] and Medica16[49]); H3K27 demethylases (GSK-J4[50]), and the H3K27 methyltransferase EZH2 (GSK-126[51]) (Supplementary Fig. 11).

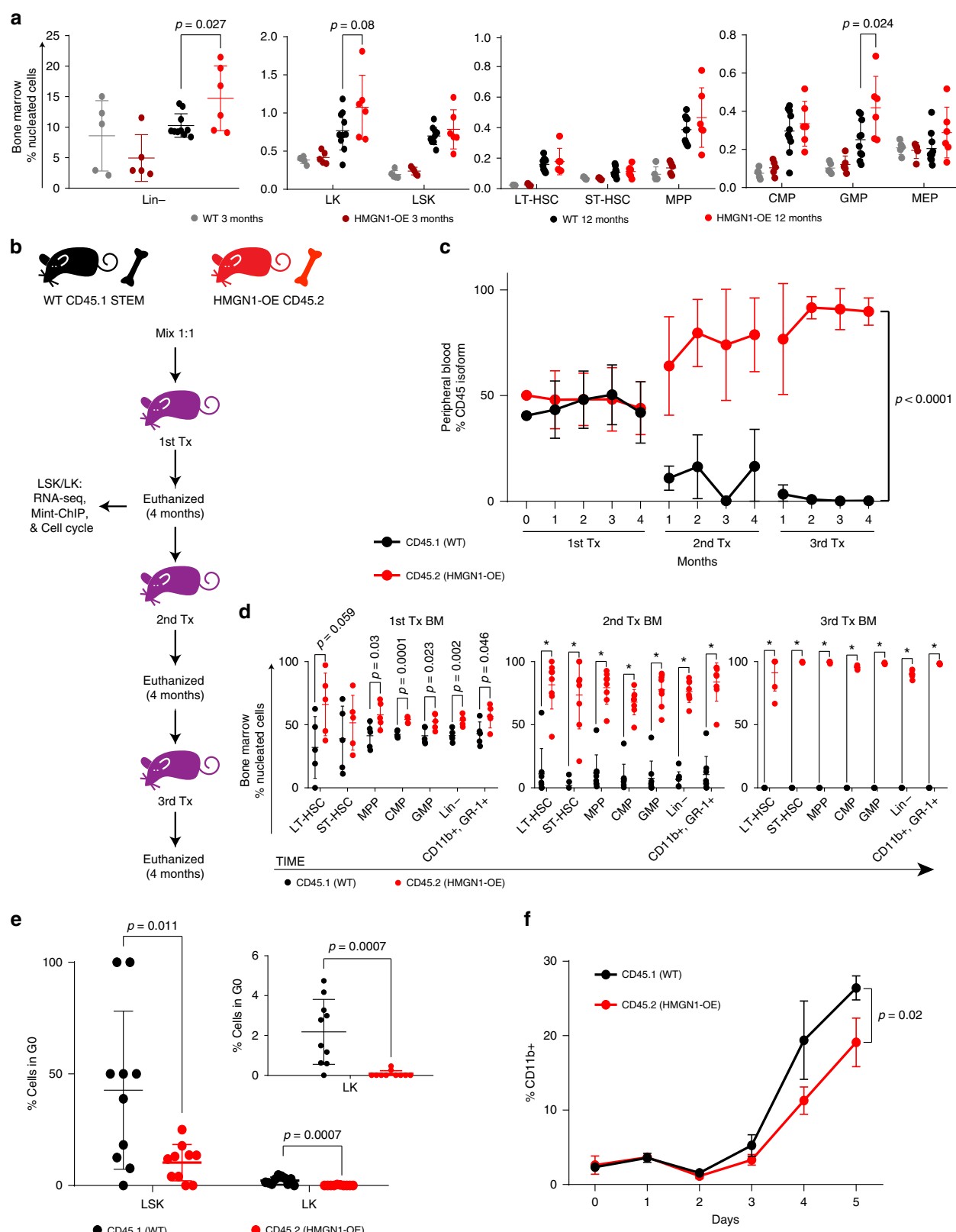

Finally, we tested HAT inhibition with C646 and A-485 in AML-ETO9a leukemia cells derived from wild-type or HMGN1-OE bone marrow. Similar to what we observed with preleukemic HSPCs (Supplementary Fig. 8b), the relative fractions of primitive LK cells were expanded and terminally differentiated GR-1+ cells

were reduced in myeloid-biased cultures of HMGN1-OE AML-ETO9a leukemia cells. Treatment with HAT inhibitors reversed the HSPC-like expansion and promoted terminal differentiation in HMGN1-OE AML-ETO9a leukemias (Fig. 6h). Together, these data suggest that HAT inhibition selectively restores

**Fig. 4 HMGN1 promotes expansion of hematopoietic precursors and provides clonal competitive advantage in vivo. a** Quantification, as percentage of total bone marrow mononuclear cells, of hematopoietic precursor subpopulations comparing young (3 months) vs. older (12 months) wild-type ($n = 5$ at 3 months, $n = 10$ at 12 months) and HMGN1-OE mice ($n = 5$ at 3 months, $n = 6$ at 12 months). **b** Schematic of the experimental approach for competitive bone marrow transplantation experiments. **c** Flow analysis of blood samples comparing CD45 isoforms from mice bled monthly after initial transplantation (first Tx, $n = 10$ mice) and subsequent serial re-transplantation (second Tx, $n = 8$ mice; and third Tx, $n = 6$ mice). **d** Flow analysis of HSPC populations in bone marrow samples 4 months after initial transplantation (first Tx, $n = 5$ independent animals) and subsequent serial re-transplantation (second Tx, $n = 8$ independent animals; and third Tx, $n = 6$ independent animals). Data are represented as comparison of CD45.1 (wild-type cells) vs. CD45.2 (HMGN1-OE cells). **e** Quantification of G0 cells within LSK and LK populations comparing wild-type and HMGN1-OE bone marrow cells in competition in vivo, $n = 10$ independent animals. **f** Analysis of CD11b by flow cytometry in CD117+ BM progenitors plated in liquid culture for 5 days ($n = 5$ independent animals). In all panels, genotypes compared by two-sided $t$ test, in panel (**d**), *$p < 0.0001$. Data are presented as mean values ± SD. Source data are provided as a Source Data file.

differentiation in HMGN1-overexpressing hematopoietic progenitors and leukemias.

## Discussion

Deregulation of the epigenome is important in transformation of hematopoietic cells, but a direct link between HMGN proteins, chromatin accessibility, and myelopoiesis has not been appreciated. We found that HMGN1 blocks differentiation, enhances stem cell properties, and cooperates with leukemia-associated oncogenes. HMGN1 overexpression increases H3K27 acetylation at loci that regulate HSCs and AML. Acetyltransferase inhibition resolved the HMGN1-associated differentiation block, suggesting that restoring normal histone marks and chromatin compaction may target leukemic phenotypes.

Polysomy 21 is highly associated with AML, but specific amplified gene(s) responsible for this association are not fully understood. Here, we linked 21q22 amplification to abnormalities in myeloid differentiation and HSPC function mediated by HMGN1. HMGN1's effect on differentiation was dependent on residues necessary for binding to the nucleosome acidic patch, which is required to modify chromatin[30]. HMGN1 increases accessible chromatin and expression-associated chromatin marks at lineage-specific loci[16,52–54], and here, similarly, we found it promoted increased expression of HSC and leukemia-associated genes in myeloid progenitors. This study adds chromatin decompaction and accessibility mediated by HMGN1 as an additional mechanism by which myeloid leukemogenesis corrupts normal hematopoietic development.

One of the striking phenotypes we observed was simultaneously decreased quiescence and increased HSC/LSC activity associated with HMGN1. These cellular states are thought to be in opposition in normal hematopoiesis, where long-term stem cells are quiescent and downstream progenitors provide the proliferative capacity to maintain blood cell production[38]. Our data suggest that chromatin changes induced by HMGN1 promote self-renewal phenotypes in a cell that also has enhanced proliferative capability. We previously observed a similar cell state where stemness and progenitor programs coexisted in individual cells from AML patients' bone marrow by single-cell RNA sequencing[55]. Future studies may identify other factors that promote this type of malignancy-associated phenotypic plasticity using the epigenomic changes induced by HMGN1 as an example.

Among the most significant targets for HMGN1 was upregulation of genes in the HoxA and HoxB clusters. Homeobox transcriptional factors in the Hox family are commonly dysregulated in AML and lead to aberrant self-renewal and development of leukemia in model systems. Overexpression of specific HoxA genes causes expansion of long-term repopulating HSCs and a myeloproliferative phenotype[56], while HoxA cluster-haploinsufficient long-term HSCs are less competitive than wild type cells in transplantation assays[57]. Our results suggest a

mechanism of upstream control of HSC and LSC activity at the chromatin. Modulating HMGN1 may have applications in expansion of normal HSCs, or conversely, in targeting LSC activity.

Clinically, these data point to the HMGN1-HAT-histone acetylation axis as a therapeutic target in AML. Furthermore, while overt DNA amplification of 21q22/*HMGN1* is observed in a subset of AMLs, the Project Achilles screen and our CRISPR knockout data suggests that even AMLs without 21q22 amplification may be susceptible to HMGN1 loss. Small molecule targeting of proteins that interact with the nucleosome acidic patch is challenging because there is no obvious pocket for drug binding, but is an active area of research[58]. In the absence of a direct inhibitor of HMGN1, HAT inhibition may be active in leukemias associated with chr21 amplification or alterations resulting in similar chromatin phenotypes.

It may be notable that while 21q22 is a recurrent target of copy number loss in solid tumors, chromosome 21 is more likely to be amplified in hematologic malignancies[59,60]. One interpretation could be that one or more elements on 21q22 are essential in the hematopoietic lineage or are tissue-specific oncogenes. Under this model, HMGN1 overexpression might be a particularly potent alteration in AML because it specifically cooperates with other events involved in transformation of blood cells. Our data suggest that HMGN1 increases chromatin accessibility to acetyltransferases such as CBP/p300, enhancing the leukemogenic activity of AML oncogenes such as t(8;21)/AML1-ETO that are known to act via histone acetylation. Further investigation is needed to determine if HMGN1-induced chromatin accessibility enhances other AML-associated oncogenes/tumor suppressor events that also act via epigenomic deregulation.

In summary, our results reveal that control of chromatin compaction may play an important role in normal hematopoiesis and preventing transformation to leukemia. HMGN1 overexpression via amplification of 21q22 impaired myeloid differentiation and increased HSC and LSC activity in vivo. HMGN1 promoted histone acetylation globally and even more so focally at leukemia-associated loci. In this context, the epigenome changes were therapeutically relevant because HMGN1 effects were reversible by inhibition of HATs. More broadly, we propose that if chromatin structure controls self-renewal and differentiation in the hematopoietic lineage, targeting chromatin accessibility may be therapeutically beneficial in AML.

## Methods

**Cell lines.** Immortalization of myeloid hematopoietic progenitors was performed by transduction of CD117 (c-Kit) positive bone marrow cells with a retrovirus expressing HoxB8 fused to the estrogen receptor[21]. Cells were grown in 10 ng/ml IL-3 (Goldbio) and 1 μM estradiol (E2) (Sigma Aldrich). AML cell lines were obtained from ATCC between 2009–2014. They were validated by STR profiling in 2019 and undergo mycoplasma testing every 6 months. Myeloid progenitors, Nomo1, and U937 cell lines were grown in RPMI 1640 supplemented with 10% fetal bovine serum (Gibco, 10438026), 1% penicillin/streptomycin (Gibco,

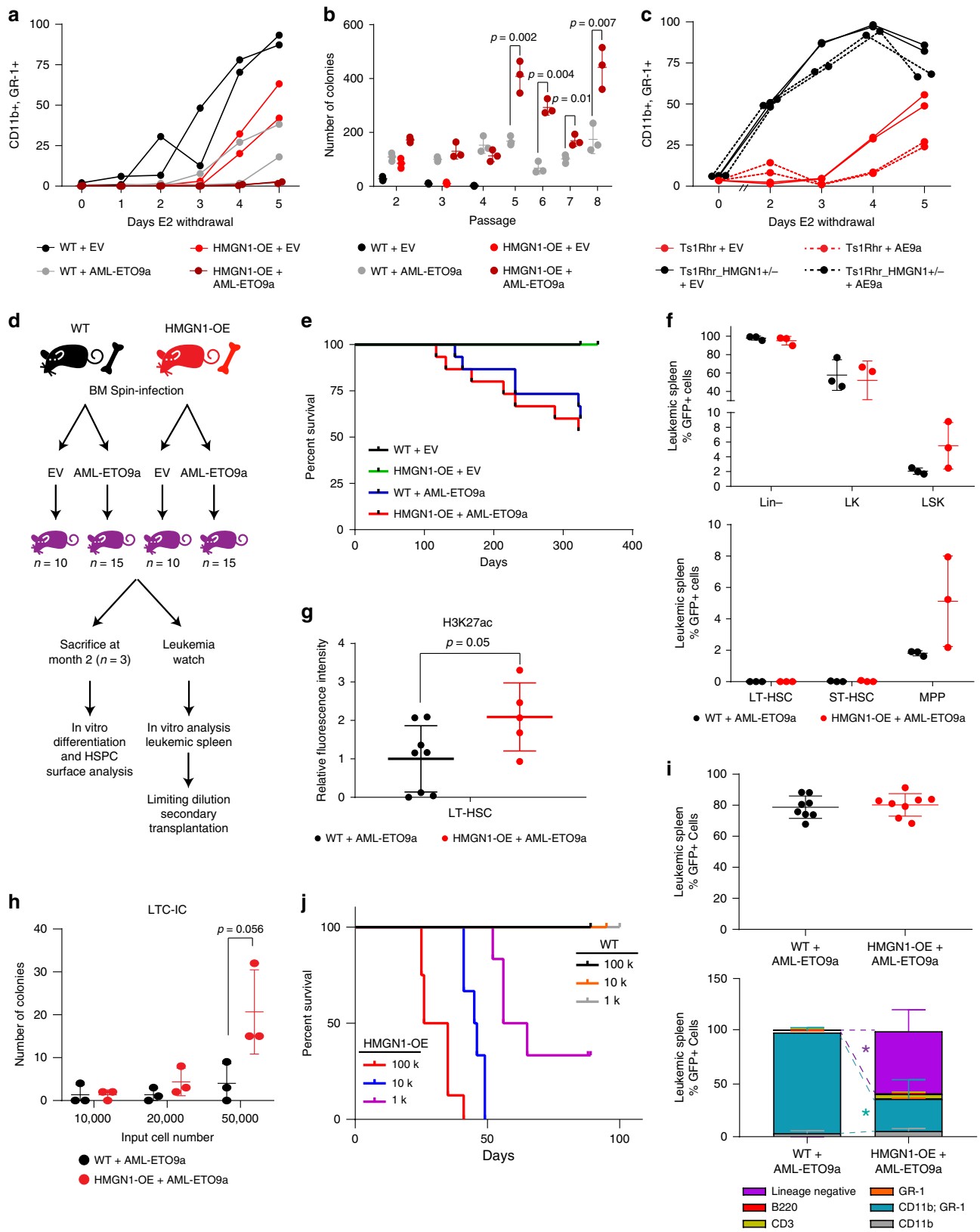

15140122), and 1% GlutaMAX (Gibco, 35050061). 293T packaging cells were grown in DMEM supplemented with 10% fetal bovine serum, 1% penicillin/streptomycin, and 1% GlutaMAX. Proliferation was measured at the indicated times starting with a concentration of 10,000 cells/ml in the case of the immortalized murine myeloid progenitors and with 100,000 cells/ml for human cell lines. Cell number was determined by a hemocytometer counting chamber.

**Mouse primary cell in vivo and ex vivo experiments**. All animal experiments were performed with approval from the Dana–Farber Cancer Institute (DFCI) Animal Care and Use Committee. All experiments were performed in a C57BL/6 (B6) background. HMGN1-OE[23] and Ts1Rhr[15] mice were back-crossed to C57BL/6 >15 generations; controls were wild-type littermates. Ts1Rhr_HMGN1[+/−] mice were generated by crossing Hmgn1[−/−] with Ts1Rhr animals[16]. For bone marrow

**Fig. 5 HMGN1 overexpression enhances LSC activity in AML-ETO9a-transformed leukemias. a** Analysis of myeloid differentiation by flow cytometry for CD11b and GR-1 in GFP+ wild-type and HMGN1-OE myeloid progenitors transduced with empty vector (EV) or AML-ETO9a-expressing retrovirus, each co-expressing GFP. $n = 2$ biologically independent cultures. **b** Serial replating assays of wild-type and HMGN1-OE cells transduced with AML-ETO9a or EV. $n = 3$ biologically independent replicates. **c** Analysis of myeloid differentiation by flow cytometry for CD11b and GR-1 in myeloid progenitors from Ts1Rhr (3 copies of Hmgn1) or Ts1Rhr_HMGN1+/− (2 copies of Hmgn1 but 3 copies of the other 30 genes in the Ts1Rhr triplication), each transduced with AML-ETO9a or EV. $n = 2$ biologically independent replicates. **d** Schematic of the experimental approach for AML-ETO9a in vivo leukemia studies. **e** Survival of primary recipients of wild-type or HMGN1-OE marrow transduced with AML-ETO9a or empty vector (EV). **f** Flow cytometry of HSPC phenotypes within GFP+ cells in spleens from moribund recipients of wild-type + AML-ETO9a or HMGN1-OE + AML-ETO9a transplants, $n = 3$ independent animals. **g** Quantification of H3K27ac levels by flow cytometry in the LT-HSC subpopulation of moribund animals' leukemias, $n = 8$ independent animals for wild-type + AML-ETO9a; $n = 5$ independent animals for HMGN1-OE + AML-ETO9a. **h** LTC-IC assays showing number of colony formation units (CFU) obtained from leukemic spleens from either AML-ETO9a wild-type or HMGN1-OE transplants, $n = 3$ independent animals. **i** Quantification of GFP+ levels and cell surface markers for subpopulations within GFP+ cells in the peripheral blood 4 weeks after secondary transplantation of 100,000 GFP+ splenocytes of the indicated genotypes, $n = 8$ independent animals. *$p < 0.0001$ comparing lineage negative or CD11b+/GR-1+ populations between genotypes. **j** Survival after limiting dilution secondary transplantation of wild-type + AML-ETO9a or HMGN1-OE + AML-ETO9a leukemia cells at the indicated cell doses (100 k $n = 8$ independent animals; 10 k $n = 6$; 1 k $n = 6$, each representing two independent leukemias of each genotype). In all cases, samples compared by two-sided $t$ test. Data are presented as mean values ± SD. Source data are provided as a Source Data file.

extraction and transplantation, mice were sacrificed, tibiae and fibulae were dissected, and cells were flushed with phosphate-buffered saline (PBS). We performed red cell lysis followed by a positive CD117 selection using magnetic beads (Miltenyi Biotec #130-094-224). For competitive transplantation, recipients were CD45.2 B6 female mice 8-9 weeks of age lethally irradiated (5.5 Gy ×2 doses) and injected with a 1:1 mixture of $10^6$ donor bone marrow cells from CD45.1(STEM) congenic wild-type mice[33] and CD45.2 OE-HMGN1 transgenic mice in a total volume of 100 μl PBS, pH 7.4. For serial transplantation, bone marrow was harvested after 16 weeks and $10^6$ cells were reinjected into the tail vein of lethally irradiated CD45.2 female recipients in a total volume of 100 μl PBS, pH 7.4.

For the in vivo assay of leukemogenic potential of HMGN1 combined with AML-ETO9a, both legs and spine from wild-type and HMGN1-OE mice were harvested by crushing and passage through a 40 μm filter. After 24 h of recovery in RPMI media enriched with IL-6 (10 ng/ml), SCF (10 ng/ml), and IL-3 (100 ng/ml), cells were infected with viral particles containing either empty pMSCV-IRES-GFP (EV) or pMSCV-AMLETO9a-IRES-GFP. Twenty-four hours post infection, $10^6$ GFP+ donor cells in a total volume of 100 μl PBS, pH 7.4 were injected into the tail vein of recipient B6 female mice 8–9 weeks of age that had been lethally irradiated (5.5 Gy ×2 doses). Animals were followed daily for sign of illness and when moribund, were sacrificed and peripheral blood, bone marrow, and splenocytes were harvested for analysis and cryopreservation.

For limiting dilution secondary transplantation, 100, 10, or 1 k GFP+ leukemic splenocytes were injected in a total volume of 100 μl PBS, pH 7.4, together with $0.5 \times 10^6$ unmanipulated wild-type bone marrow cells into the tail vein of recipient B6 female mice 8–9 weeks of age that had been lethally irradiated (5.5 Gy ×2 doses). For each condition, cells from 2 different leukemic spleens were used and each were injected into 4 (for the 100 k condition), 3 (10 k), and 3 (1 k) recipients. LSC frequency in limiting dilution transplants was calculated using L-Calc software (StemSoft). For in vitro culture and differentiation of primary hematopoietic progenitors, with or without drug treatment, fresh bone marrow cells, CD117+ enriched, were seeded in RPMI media containing 10% FBS and IL-6 (10 ng/ml), SCF (10 ng/ml), and IL-3 (100 ng/ml). The same culture conditions were used for ex vivo culture of CD117+ enriched cells after transduction with oncogene-containing retroviruses co-expressing GFP.

**Clonogenic assays.** Wild-type or HMGN1-OE cells stably infected with the oncogenes indicated were seeded in methylcellulose media (Methocult M3234, Stem Cell Technologies), supplemented with IL-6 (10 ng/ml), SCF (10 ng/ml), and IL-3 (6 ng/ml) (GoldBio) at $2 \times 10^5$ cells/ml and at $5 \times 10^4$ cells/ml in subsequent passages. Colonies were manually counted at 7 days, pooled, and replated. For long-term culture initiating colony (LTC-IC) assays, leukemic splenocytes from three different animals were plated on MS5 feeder cells for three weeks in RPMI media enriched with IL-6 (10 ng/ml), SCF (10 ng/ml), and IL-3 (6 ng/ml). Cells were counted and 10,000, 20,000, or 50,000 GFP+ cells were seeded in methylcellulose media (Methocult M3234) supplemented with SCF (10 ng/ml), IL-3 (6 ng/ml), and IL-6 (10 ng/ml) in 35 mm dishes. One week later, the number of colonies and cells was determined.

**Antibodies.** Detailed information of antibodies used for Western blotting, flow cytometry, Mint-ChIP, and ChIP-PCR is included in Table 1.

**Drug treatment assays.** Cell lines were plated in a 96-well dish at a concentration of 2000 cells per 160 μl of media with drugs or vehicle (DMSO). Compounds were added to the cells in a serial threefold dilution. After incubation for 72 h at 37 °C, viability was determined by a 3-(4.5-dimehtylthiazol-2-yl)-2,5-diphenyltetrazolium bromide (MTT, Sigma) assay according to manufacturer's instructions. Briefly, 20 μl of 5 mg/ml MTT was added to each well and incubated for 2 h at 37 °C followed

by addition of 100 μl of MTT lysis buffer and overnight incubation at 37 °C. Absorbance values were measured using a SpectraMax M3 plate reader at 570 and 630 nm. Viability curve values were analyzed by GraphPad Prism 8 (GraphPad Software, Inc.) by nonlinear regression analysis. Detailed information for the compounds is in Table 2.

**Flow cytometry.** For in vitro differentiation, cells were treated with compounds or vehicle at doses and times indicated. Cells were harvested and washed with PBS before staining and then incubated with the antibodies indicated for 30 min in the dark at 4 °C followed by a final wash in PBS before flow cytometry analysis. In the case of murine peripheral blood samples, we initially lysed the samples twice with red cells lysis buffer prior to staining. For analysis of bone marrow or spleen samples, samples were lysed once. Antibodies for staining are indicated in Table 1. Gating for HSPC subpopulations were defined as indicated in Table 3.

For analysis of H3K27ac levels by flow cytometry, cells were harvested and washed twice with PBS followed by fixation with 4% paraformaldehyde in PBS for 15 min on ice. Cells were pelleted by centrifugation for 5 min at 300g and resuspended in cold 70% EtOH added drop by drop while vortexing and then incubated overnight at −20 °C. Next, cells were washed twice with PBS and once with PBS/1% FBS. Cells were permeabilized for 20 min with 0.25% Triton-X100 in PBS/1% FBS followed by incubation with primary at 1:500 dilution for 30 min and then secondary antibody at 1:1000 dilution for 2 h at 4 °C in the dark. For analysis of cell cycle distribution, cells were initially stained with surface markers as described above and then permeabilized and fixed as indicated for H3K27ac analysis above. Next, cells were washed twice with PBS/2% FBS and stained with 1 μM Hoechst 33342 in HBSS (Hank's balanced salt solution) supplemented with 20 mM HEPES pH 8 and 10% FBS. After 30 min of incubation at 37 °C, Pyronin Y was added to a final concentration of 1 μg/ml incubated for an additional 30-minutes. Cells were analyzed using a CytoFlex flow cytometer (Beckman Coulter) in most cases. For analysis of HSPCs, cells were analyzed using a FACS Canto II. In all cases, data was processed using FlowJo X software. For sorting of LK and LSK progenitor populations, cells were incubated with the antibodies indicated on each case and sorter in a FACS Aria II SORP. Table 1 includes complete information on antibodies.

**Cellular ROS measurement.** For measurement of ROS production during differentiation, we used the cell permeable reagent 2′,7″-dichlorofluorescein diacetate (DCFDA, Life technologies D399) following the manufacturer's instructions. Briefly, cells were harvested and incubated with 25 μM DFCDA in culture media for 30 min at 37 °C and then analyzed by flow cytometry for fluorescence shift caused by DFCDA oxidation to DCF.

**Cytospins for cellular morphology.** Twenty thousand cells were washed and resuspended in 100 μl of PBS and centrifugation was performed at 10×g for 3 min on a CytoSpin 3 centrifuge (Shandon). Slides were air-dried and then stained for 5 min in May-Grunwald solution followed by a 2 min wash in PBS. Next, slides were stained for 15 min in a 1:20 dilution in Giemsa and washed briefly 2–3 times in $H_2O$. Slides were air-dried before visualization.

**Viral infections.** Lentiviral and retroviral particles were produced in 293T cells using standard procedures with Lipofectamine 2000 (Invitrogen, 11668027). For retroviral production, 293T cells in a 10 cm plate were transfected with 12 μg of pECO-PAC together with 12 μg of the viral construct of interest and viral supernatant was harvested 48 h later. In the case of lentivirus production, 293T cells were transfected with 10.8 μg pPAX2, 2.4 μg pVSVg, and 10.8 μg of the viral expression plasmid of interest. Viral supernatant was harvested 48- and 72-h post transfection and concentrated by ultracentrifugation at 23,000g for 2 h at 4 °C. For infection,

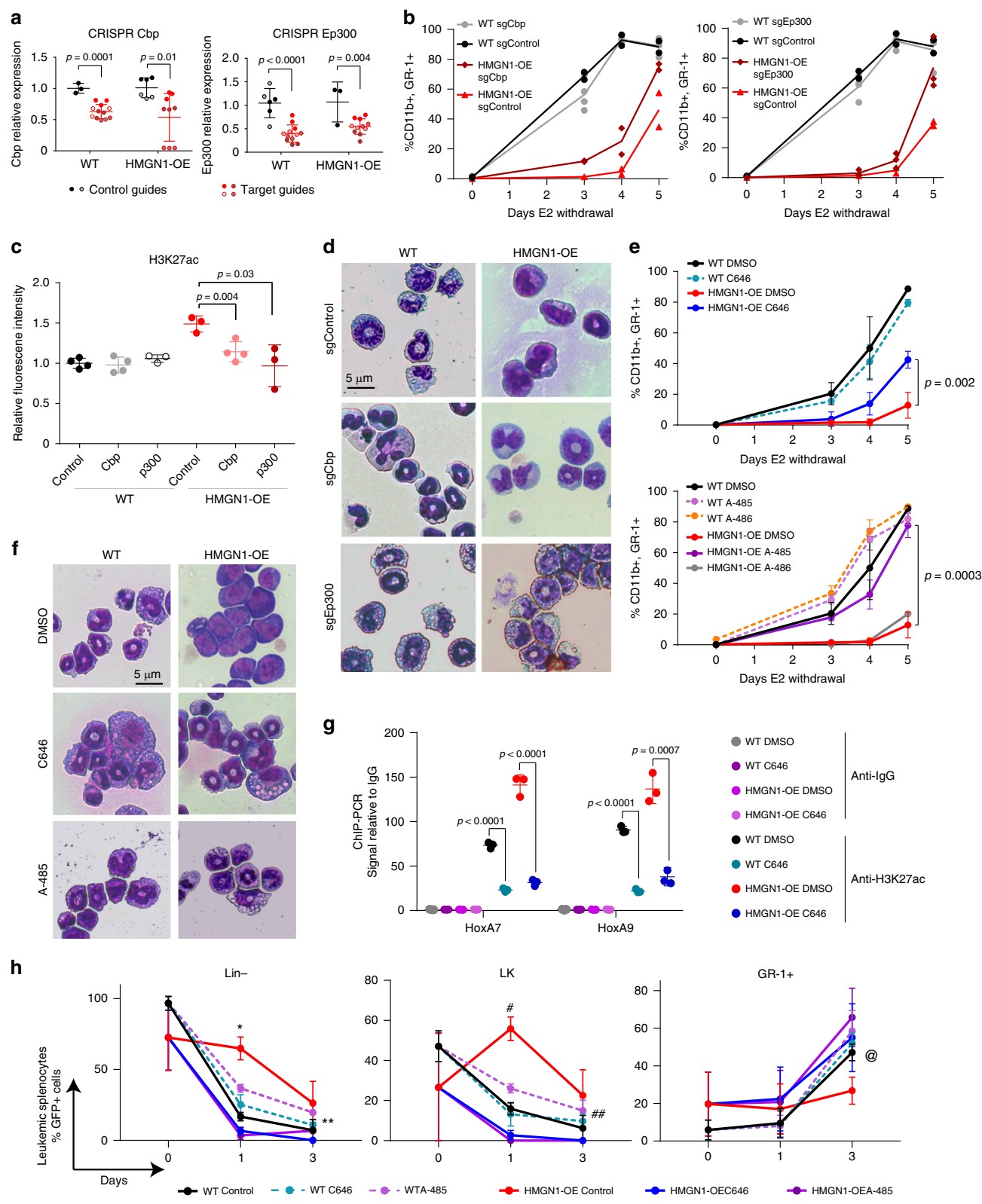

200,000 actively dividing cells were spinfected in the presence of 1.5 ml of viral suspension and polybrene (final concentration 8 μg/ml) at 2000g for 90 min at 32 °C. To knock-down *Hmgn1* in stably infected cells, shRNAs were induced with 0.5 μg/ml doxycycline for 96 h before performing assays.

**cDNA and shRNA expression and CRISPR/Cas9 gene targeting.** For constitutive expression of HMGN1-WT and HMGN1-SEmut (mutant protein unable to bind to nucleosomes[16]) we used a pMSCV backbone, adding an in-frame FLAG

tag using the primers indicated in Table 4. For modulation of HMGN1 expression in Ts1Rhr cells by shRNA, we used the doxycycline-inducible tet on-pLKO system targeting the sequences indicated in Table 4. For the CRISPR–Cas9 approach, we initially generated Flag–Cas9 stable cell lines by lentiviral infection of pCRISPRV2-FLAG-CAS9 (Addgene #52961) as described above, followed by selection with 2 μg/ml puromycin. Next, for targeting of either *HMGN1* (human AML cell lines), or *Ep300* or *Cbp* (murine myeloid progenitors) we used the pLKO5-RFP657 vector (Addgene #57824). Oligonucleotide sequences are in Table 4.

**Fig. 6 Targeting histone acetyltransferases CBP/p300 reverses HMGN1-associated myeloid differentiation abnormalities. a** Q-RT-PCR of *Cbp* and *Ep300* expression after infection with CRISPR guides targeting each gene (*n* = 4 biologically independent guides) or control non-targeting guides (*n* = 2 biologically independent guides). **b** Effects of sgRNAs targeting *Cbp* (left panel) or *Ep300* (right panel) compared to controls upon differentiation of wild-type and HMGN1-OE progenitors after E2 withdrawal. Lines are mean of biologically independent guide-expressing cultures (WT and OE sgControl = 2 guides, WT sgCbp and sgEp300 *n* = 2 guides each in biological duplicate, OE sgCbp = 2 guides, OE sgEp300 = 2 guides, one in biological duplicate). **c** Flow cytometry analysis of H3K27ac in wild-type and HMGN1-OE myeloid progenitors expressing sgRNAs targeting *Cbp* and *Ep300* compared to controls. *n* = 3 or 4 biologically independent samples, as indicated. **d** Morphological assessment of *Cbp* and *Ep300* sgRNA effects on myeloid differentiation 4 days after E2 withdrawal. Data are representative of 4 biologically independent experiments. **e** Effects of HAT inhibitors 0.5 μM C646 and 0.5 μM A-485 (compared to inactive analog A-486) on differentiation after withdrawal of E2 measured by flow cytometry for CD11b and GR-1. *n* = 3 biologically independent samples. **f** Morphologic assessment of C646 and A-485 effects on myeloid differentiation. Data are representative of three biologically independent experiments. **g** ChIP-PCR in wild-type and HMGN1-OE myeloid progenitors after 96 h of E2 withdrawal with or without C646 for relative H3K27ac at the *HoxA7* and *HoxA9* loci. *n* = 3 biologically independent samples. **h** Effects of HAT inhibitors 0.5 μM C646 and 0.5 μM A-485 on proportion of the indicated subpopulations of AML-ETO9a leukemic splenocytes derived from HMGN1-OE or WT progenitors. *n* = 3 biologic replicates. *$p$ = 0.003 OE control vs. C646, $p$ = 0.002 OE control vs. A-485; **$p$ = 0.04 OE control vs. C646; #$p$ = 0.0001 OE control vs. C646, $p$ = 0.0001 OE control vs. A-485; ##$p$ = 0.03 OE control vs. C646, $p$ = 0.03 OE control vs. A-485; @$p$ = 0.06 OE control vs. C646, $p$ = 0.01 OE control vs. A-485. In all panels, samples compared by two-sided *t* test. Data are presented as mean values ± SD. Source data are provided as a Source Data file.

**RNA sequencing**. Murine myeloid progenitor cells were stimulated to differentiate for the indicated times. One million cells were counted, and RNA was extracted with TRIzol (Invitrogen, 15596018). For per cell normalization, 1 μl of Mix #1 ERCC exogenous spike-in RNA (Ambion, 4456740, diluted 1:1000) was added to each RNA sample. In the case of competitive bone marrow transplantations, LK and LSK populations were sorted by pooling cells from three different biological mice in triplicate per sample (nine mice in total per genotype per timepoint). RNA quality checks were performed using the RNA Qubit Assay (Invitrogen) and on a Bioanalyzer RNA Nano 6000 Chip Kit (Agilent). Libraries were prepared using the TruSeq RNA Library Preparation Kit v2, Set A (Illumina, RS-122-2001). Sequencing was performed on a NextSeq500 (Illumina) by the Molecular Biology Core Facilities (MBCF) at DFCI. RNA-seq data processing included normalization to spike-in controls where indicated[25,61]. Bowtie (version 0.12.2) was used to align sequences to a genome build that included ERCC synthetic spike-in RNA sequences (https://tools.thermofisher.com/content/sfs/manuals/cms_095047.txt). For each gene and spike-in RNA, we computed a value for reads per kilobase of transcript, per million mapped reads (RPKM). Loess regression was used to normalize RPKM values using the spike-in values as reference by performing regression on the combined data matrix with *loess.normalize* in the R package *affy*. Here, *mat* defined the total RPKM matrix and *subset* (for normalization) were the ERCC spike-ins. Other parameters were set to default values.

**Gene set enrichment analysis (GSEA)**. GSEA (http://www.broadinstitute.org/gsea/)[15] was performed using the Molecular Signatures Database (MSigDB) version 6.0[62]. Leading edge analysis was performed in by GSEA and visualized as heat maps using Morpheus (http://software.broadinstitute.org/morpheus/).

**Chromatin immunoprecipitation and sequencing (Mint-ChIP)**. For ChIP-seq on myeloid progenitors and on LSK and LK cells, we employed Mint-ChIP[26] with some modifications. Briefly, cells were lysed, followed by MNase digestion of the chromatin and barcoded adapter ligation. Samples were pooled and total H3, H3K27ac, and H3K27me3 antibodies (see Table 1) were used for immunoprecipitation. In vitro transcription, reverse transcription, and PCR were used for library generation. Compared to the original Mint-ChIP protocol, changes in adapter and oligo designs were made to place the adapter barcode adjacent to SBS12 instead of SBS3. Details of the protocol are described at https://www.protocols.io/view/mint-chip3-a-low-input-chip-seq-protocol-using-mul-wbefaje. Gene tracks were visualized in the Integrated Genomics Viewer (IGV, Broad Institute).

**Quantitative proteomics (Tandem mass tag analysis)**. Myeloid progenitor cells were washed twice with ice-cold PBS and lysed in urea buffer (8 M urea, 75 mM NaCl, 50 mM EPPS pH 8.0, complete protease inhibitors (Roche) and PhosSTOP phosphatase inhibitors (Roche). Protein concentration was determined by bicinchoninic acid (Thermo Fisher Scientific). Next, 100 μg of protein were reduced with 5 mM TCEP and alkylated with 14 mM iodoacetamide (IAA). All incubations were performed at room temperature for 30 min. Proteins were precipitated by methanol/chloroform, resuspend in 200 mM EPPS and digested with LysC for 12 h at room temperature (1:100, LysC:Protein). Then, trypsin was added to the peptide mixture and further digested for 5 h at 37 °C (1:75, Trypsin:Protein). After digestion, peptide concentration was calculated using the quantitative colorimetric peptide assay (Pierce). A total of 25 μg of peptides were labeled with TMT-10plex[63]. After labeling, all samples were combined in equal amounts and fractionated in a basic pH reverse phase chromatography. All 96 fractions collected were combined into 24, 12 of which were desalted via STAGE-TiP[64] and dried in a vacuum centrifuge. Finally, peptides were resuspended in 3% ACN, 1% formic acid and analyzed by LC-SPS-MS3[65] in an Orbitrap Fusion (Thermo Fisher Scientific)

coupled to a Proxeon EASY-nLC II LC pump (Thermo Fisher Scientific). Peptides were fractionated and identified mass spectra were searched against the human Uniprot database; differentially expressed proteins were identified by fitting to a linear model[66,67].

**Western blotting**. Samples for Western blotting were prepared by lysing 10^6 cells in radioimmunoprecipitation assay buffer containing 1× protease inhibitor. The antibodies used are detailed in Table 1. Blots were imaged using an ImageQuant LAS-4000.

**ChIP-PCR**. ChIP-PCR was performed using the SimpleChIP enzymatic chromatin IP kit (Cell Signaling Technology, #9005) per the manufacturer's instructions.

**ATAC-seq**. An Assay for Transposase-Accessible Chromatin using sequencing (ATAC-seq) was performed using the Omni-ATAC protocol[68]. Briefly, $10^5$ cells were resuspended in 1 ml of cold ATAC-seq resuspension buffer (RSB; 10 mM Tris-HCl pH 7.4, 10 mM NaCl and 3 mM MgCl₂ in water). Cells were centrifuged at 500×g for 5 min in a prechilled fixed-angle centrifuge and the supernatant was carefully aspirated. Cell pellets were then resuspended in 50 μl RSB containing 0.1% NP40, 0.1% Tween-20 and 0.01% digitonin y pipetting up and down three times. This cell lysis reaction was incubated on ice for 3 min. After lysis, 1 ml RSB containing 0.1% Tween-20 (without NP40 or digitonin) was added, and the tubes were inverted to mix. Nuclei were then centrifuged for 10 min at 500×g in a prechilled fixed-angle centrifuge. Supernatant was removed, and nuclei were resuspended in 50 μl of transposition mix (2.5 μl transposase (100 nM final), 16.5 μl PBS, 0.5 μl 1% digitonin, 0.5 μl 10% Tween-10 and 5 μl water) by pipetting up and down six times. Transposition reactions were incubated at 37 °C for 30 min in a shaking thermo-mixer. Reaction were cleaned up with QIAquick PCR spin columns. Library quantitation was used to determine the number of amplification cycles by plotting linear Rn vs. cycle and determining the cycle number corresponding to one-fourth of maximum fluorescent intensity[69]. After sequencing on a NextSeq500 per the manufacturer's instructions, we used ChiLin pipeline 2.0.0 for QC and pre-processing[70], Burrows–Wheeler Aligner for read mapping[71], Model-based Analysis of ChIP-Seq (MACS) as a peak caller[72], and DESeq2 for differential peak analysis[73].

**Quantitative RT-PCR**. RNA was prepared using the Trizol reagent, cDNA was prepared using the High-Capacity cDNA Reverse Transcription Kit (Applied Biosystems, 4368814), and quantitative PCR was performed using the Power SYBR Green PCR Master Mix (Applied Biosystems, 4367359), all per the manufacturers' instructions.

**Statistical analysis**. Prism software (GraphPad) was used for calculating statistical significance, except as where noted below. Plots of colony numbers, differentiation profiles, surface isotyping, H3K27 acetylation, RNA expression by quantitative RT-PCR, and ChIP-PCR, that contain summary statistics represent the mean of at least three replicates and error bars are SD. Statistical significance was compared for groups with at least three biologically independent replicates by unpaired two-tailed *t* test without multiple hypothesis correction, except as where indicated. GSEA was performed using 1000 permutations by gene set, weighted enrichment statistic, genes ranked by log2 ratio. The GSEA enrichment score reflects the degree to which a gene set is overrepresented at the upper or lower ends of a ranked list of genes. A running-sum statistic is generated from the ranked list of genes, with the magnitude of the increment depending on the correlation of the gene with the phenotype. The enrichment score is the maximum deviation from zero. The false-

**Table 1 List of antibodies.**

| Target-fluorophore | Provider-catalog number | Experiment | Dilution | Target | Provider-catalog number | Experiment | Dilution |
|---|---|---|---|---|---|---|---|
| CD11b-PE | Life Technologies #12-0112-81 | Flow Cytometry | 1:500 | HMGN1 | Life Technologies #A302363A | Western Blot | 1:1000 |
| GR1-FITC | Fisher Scientific #509919 | Flow Cytometry | 1:500 | B-Actin | Sigma, #SAB5500001 | Western Blot | 1:10,000 |
| Anti-rabbit Alexa555 | Life Technologies #A21428 | Flow Cytometry | 1:1000 | Histone H3 | Cell Signaling Technology #9715 | Western Blot | 1:1000 |
| Lineage Cell Detection Cocktail-Biotin, mouse | Miltenyi Biotech #130-092-613 | Flow Cytometry | 1:10 | Histone H3 K27ac | Cell Signaling Technology 8173 | Western Blot and Flow Cytometry | 1:1000 |
| Streptavidin-PE/Cy7 | BioLegend # 405233 | Flow Cytometry | 1:100 | Histone H3 K27ac | Abcam, #ab4729 | ChIP-PCR/WB/FC | 1:100 (ChIP & Flow Cytometry); 1:1000 (WB) |
| CD117-APC | Fisher Scientific #BDB553356 | Flow Cytometry | 1:100 | Histone H3 K27me3 | Active Motif #61017 | ChIP-PCR | 1:100 |
| CD16/32 APC/Cy7 | Fisher Scientific #BDB560541 | Flow Cytometry | 1:100 | CBP | Cell Signaling Technology #7425 | ChIP-PCR | 1:100 |
| CD34-Alexa700 | Fisher Scientific #BDB560518 | Flow Cytometry | 1:100 | p300 | Bethyl Laboratories #A300-358A-M | ChIP-PCR | 1:100 |
| CD150-Pac Blue | BioLegend #115924 | Flow Cytometry | 1:100 | Histone H3 | Active Motif #39763 | Mint-ChIP | 1:100 |
| CD48-PE/Cy5 | BioLegend #103420 | Flow Cytometry | 1:100 | Histone H3 K27ac | Active Motif #39133 | Mint-ChIP | 1:100 |
| CD45.1-FITC | BioLegend #110706 | Flow Cytometry | 1:100 | Histone H3 K27me3 | Millipore #07-449 | Mint-ChIP | 1:100 |
| CD45.2-PERCP/Cy5.5 | BioLegend #109828 | Flow Cytometry | 1:100 | | | | |
| B220-PE | BioLegend #103207 | Flow Cytometry | 1:100 | | | | |
| CD3-PB | Fisher Scientific #BDB558214 | Flow Cytometry | 1:100 | | | | |
| GR1-APC | BioLegend #108412 | Flow Cytometry | 1:100 | | | | |
| CD11b-APC/CY7 | BioLegend-#101226 | Flow Cytometry | 1:100 | | | | |

**Table 2 List of compounds and other reagents.**

| Reagent | Catalog number-provider | Reagent | Catalog number-provider |
|---|---|---|---|
| Estradiol | Sigma-Aldrich #E2758 | Medica16 | Sigma-Aldrich #M5693 |
| IL-3 | Gold Bio technology #1310-06-10 | GSK-J4 | Selleck #7070 |
| IL-6 | Gold Bio technology | GSK-126 | Selleck #S7061 |
| C646 | Selleck #S7152 | Doxycycline hydrochloride | Fisher Scientific #BP26531 |
| A-485 and A-486 | Structural Genomics Consortium | Trizol | Life Technologies #15596018 |
| JQ1 | Selleck #S7110 | ERCC Spike-in | Thermo Fisher/Ambion #4456740 |
| SCF | Gold Bio technology #1320-01-10 | Puromycin dihydrochloride | Gold biotechnology #P-600 |
| Cbp30 | Sigma-Aldrich #SML1133 | Polybrene | Santa Cruz #sc-134220 |
| iCbp112 | Sigma-Aldrich #SML1134 | Lipofectamine2000 | Life technologies #11668500 |
| BMS-3031411 | Sigma-Aldrich #SML0784 | RNAse | Sigma-Aldrich #R6513 |
| Hoechst33342 | Sigma-Aldrich#B2261 | PyroninY | Sigma-Aldrich #P9172 |
| DCFDA | Life technologies #D399 | Power SYBR Green PCR Master mix | Applied Biosystems #4367359 |
| Qubit RNA BR assay kit | Life Technologies #Q10210 | Bioanalyzer RNA 600 Chip Kit | Agilent #5067-1511 |
| RIPA | Boston Bioproducts #BP-115 | Protease inhibitor | ThermoFisher #87786 |
| High-capacity cDNA Reverse Transcription Kit | Applied Biosystems #4368814 | SimpleChIP enzymatic chromatin IP kit | Cell Signaling Technology #9005 |

**Table 3 Markers of hematopoietic subpopulations.**

| HSPC population | Surface markers for flow cytometry |
|---|---|
| LSK | Lin−, CD117+, Sca-1+ |
| LT-HSC (long-term HSC) | Lin−, CD117+, Sca-1+, CD150+, CD48− |
| ST-HSC (short-term HSC) | Lin−, CD117+, Sca-1+, CD150−, CD48− |
| MPP (multipotent progenitor) | Lin−, CD117+, Sca-1+, CD150−, CD48+ |
| LK | Lin−, CD117+, Sca-1- |
| GMP (granulocyte–monocyte progenitor) | Lin−, CD117+, Sca-1-, CD34+, CD16/32+ |
| CMP (common myeloid progenitor) | Lin−, CD117+, Sca-1-, CD34+, CD16/32− |
| MEP (megakaryocyte–erythroid progenitor) | Lin−, CD117+, Sca-1-, CD34−, CD16/32− |

**Table 4 List of primers.**

| Target | Sense sequence 5′-3′ | Anti-sense sequence 5′-3′ | Experiment |
|---|---|---|---|
| HMGN1 #1 | TTCTATCTGGTCCCGTGTTTC | GAAACACGGGACCAGATAGAA | shHMGN1 |
| HMGN1 #2 | TGTGGTCATGGCAGTCCATTT | AAATGGACTGCCATGACCTCA | shHMGN1 |
| Flag-HMGN1 | GGGGACAACTTTGTACAAAAAAGTT | GGGGACAACTTTGTACAAGAAAGTT | Flag-HMGN1 |
|  | GGCATGCCCAAGAGGAAGGTTAGCGC | GCTACTTGTCGTCATCGTCTTTGTAG | |
|  |  | TCGTCGGACTTAGCTTCTTTCTCTTC | |
| Control_g1 | CACCGACGGAGGCTAAGCGTCGCAA | AAACTTGCGACGTTAGCCTCCGTC | CRISPR (Control) |
| Control_g2 | CACCGCGCTTCCGCGGCCCGTTCAA | AAACTTGAACGGGCCGCGGAAGCGC | CRISPR (Control) |
| HMGN1_g1 | CACCGCCGCAGGTCAGCTCCGCCGA | AAACTCGGCGGAGCTGACCTGCGGC | CRISPR (HMGN1) |
| HMGN1_g2 | CACCGTTCGTTTCCCCGTTTTCCGC | AAACGCGGAAAACGGGGAAACGAAC | CRISPR (HMGN1) |
| Cbp_g1 | CACCGCGTGTATACATATCTTATC | AAACGATAAGATATGTATACACGC | CRISPR (Cbp) |
| Cbp_g2 | CACCGCCCCAAACCAAAACGACTAC | AAACGTAGTCGTTTTGGTTTGGGGC | CRISPR (Cbp) |
| Cbp_g3 | CACCGTCAAACAAGCGAACGAAGAC | AAACGTCTTCGTTCGCTTGTTTGAC | CRISPR (Cbp) |
| Cbp_g4 | CACCGCAGCTTCTGCGACAGGTCGT | AAACACGACCTGTCGCAGAAGCTGC | CRISPR (Cbp) |
| Ep300_g1 | CACCGAACAGGGCCTTTGTTCGGTA | AAACTACCGAACAAAGGCCCTGTTC | CRISPR (p300) |
| Ep300_g2 | CACCGCAGTCCGCAAGCATTTAGGA | AAACTCCTAAATGCTTGCGGACTGC | CRISPR (p300) |
| Ep300_g3 | CACCGTACCATTCTTGCAGGCGCT | AAACAGCGCCTGCAAGAATGGTAC | CRISPR (p300) |
| Ep300_g4 | CACCGTTAGACACATTGGGCATACC | AAACGGTATGCCCAATGTGTCTAAC | CRISPR (p300) |
| HoxA7 | TATGTGAACGCGCTTTTTAGCA | GGGGGCTGTTGACATTGTATAA | Q-RT-PCR |
| HoxA9 | CCCCGACTTCAGTCCTTGC | GATGCACGTAGGGGTGGTG | Q-RT-PCR |
| Cbp | TTCTCCGCGAATGACAACACA | CCTGGGTTGATGCTAGAGCC | Q-RT-PCR |
| p300 | AATGGACAAGGGATAATGCCCA | CTCAGTCAATAAACTGCCAGCA | Q-RT-PCR |
| HoxA7 | GCCACAACCCCTAGTTACCC | GGAGCCGAGTTTCTCCCCAAA | ChIP-PCR |
| HoxA9 | CCACGCTTGACACTCACAC | TCGGCATTGTTTTCGGAGAAG | ChIP-PCR |

discovery rate in GSEA is the estimated probability that a gene set with a given enrichment score represents a false positive and is calculated as a ratio of two distributions: (1) the actual enrichment score vs. the enrichment score for all gene sets against all permutations of the dataset, and (2) the actual enrichment score versus the enrichment score for all gene sets against the actual dataset.

**Reporting summary**. Further information on research design is available in the Nature Research Reporting Summary linked to this article.

## Data availability
RNA-seq, ATAC-seq, and Mint-ChIP data have been deposited in the GEO database under the SuperSeries accession code GSE143683. RNA-seq is in SubSeries GSE142473, ATAC-seq is in SubSeries GSA143840, and Mint-ChIP is in SubSeries GSE143682. Mass spectrometry proteomic data are available via ProteomeXchange with identifier PXD017054. All the other data supporting the findings of this study are available within the article and its Supplementary information files and from the corresponding author upon reasonable request. The source data underlying Figs. 1–6 and Supplementary Figs. 1–3, 5, and 7–11 are provided as a Source Data file. A reporting summary for this article is available as a Supplementary Information file.

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

## Acknowledgements
P.v.G. was supported by the Leukemia & Lymphoma Society and National Cancer Institute grant CA218832. A.A.L. was supported by National Cancer Institute grants CA181340 and CA225191, Alex's Lemonade Stand Foundation, Leukemia & Lymphoma Society, Harvard Stem Cell Institute, the Anna Fuller Fund, AIM 2 Cure, and Curing Kids Cancer. The authors thank Francois Mercier and David Scadden for sharing the CD45.1(STEM) mice, and Zach Herbert and the DFCI Molecular Biology Core Facility for assistance with sequencing. Cell cartoons in Fig. 2a are open source via CC BY 3.0 (https://creativecommons.org/licenses/by/3.0/) from http://smart.servier.com. A-485 and A-486 were supplied by the Structural Genomics Consortium under an Open Science Trust Agreement: http://www.thesgc.org/click-trust.

## Author contributions
L.C.H., P.v.G., M.A.P., K.J.H., K.T., C.T.M., J.A.P., Y.X., P.C., H.W.L., D.B.S., S.P.G., D.J.F., B.E.B., and A.A.L. performed the experiments and analyzed the data; T.F. and M.B. provided essential reagents and analyzed the data; L.C.H. and A.A.L. designed the project and wrote the paper; all authors edited the paper.

## Competing interests
A.A.L. has received research funding from AbbVie and Stemline Therapeutics and serves as a consultant for N-of-One/Qiagen.
