## [Peer Review File · Nature Communications]

REVIEWERS' COMMENTS:

Reviewer #2 (Remarks to the Author):

The authors have addressed the reviewer's comments.

Reviewer #3 (Remarks to the Author):

Accept.

Reviewer #4 (Remarks to the Author):

In their revised manuscript, Cabal-Hierro et al provide extensive new data that thoroughly address the reviewers' points. These new data include genomic analyses to examine chromatin accessibility and changes in both activating and repressive H3K27 modifications upon HMGN1 overexpression. Furthermore, multiple models of HMGN1 overexpression are used in cell culture and in vivo, and both loss of function and gain of function experiments were performed in multiple models to validate the key observations. Although I did not see the initial submission, the manuscript appears to be considerably improved by these additions. Overall, I find the work to be of broad interest, the data of high quality, and the manuscript to be well written.

Below, I have a few minor comments for the authors' consideration.

1. In several panels, heatmaps are plotted as row minima and maxima (Fig. 2b, c, d, f; Fig. 3d). This representation obscures the actual differences among samples, which may be high for some genes and low for others. The authors should consider plotting these as $\log_2(\text{fold change})$ values relative to a common reference such as WT_0h.

2. In the Fig. 2e metagene plots, are the data aggregated over all ATAC-seq peaks of either genotype, all differential peaks (both increased and decreased in OE cells), or only peaks increased in OE cells? If only differential peaks or increased peaks are plotted, I suggest showing two metagene plots: one for all peaks (which may show minimal overall changes in accessibility) and one for differential peaks.

3. Fig. 2d suggests that even in OE cells, Hox genes are silenced after 96 hours of differentiation. Therefore, the differentiation block in these cells is likely due to the combined effects of numerous gene expression changes. The authors should comment on this more explicitly.

4. In Fig. 3b or the figure legend, the authors should specify what ChIP-seq data is being depicted. From context, it appears that genes with different H3K27ac levels are examined here, but this should be stated directly since both H3K27ac and H3K27me3 profiling were performed.

5. P14: "...(LTC-IC) assays, an in vitro measure of HSPC activity that also correlates with poorer risk in human AML41..." Does "poorer risk" mean "higher risk" and/or "poorer survival"? This term seems unnecessarily confusing.

Response to Reviewers for Cabal-Hierro, et al.

We thank the Reviewers for their positive opinion of our manuscript. We have addressed the points raised by Reviewer 4, as outlined below. The original Reviewer remarks are in black Times font and our responses are in red Arial font.

REVIEWERS' COMMENTS:

Reviewer #2 (Remarks to the Author):

The authors have addressed the reviewer's comments.

Reviewer #3 (Remarks to the Author):

Accept.

Reviewer #4 (Remarks to the Author):

In their revised manuscript, Cabal-Hierro et al provide extensive new data that thoroughly address the reviewers' points. These new data include genomic analyses to examine chromatin accessibility and changes in both activating and repressive H3K27 modifications upon HMGN1 overexpression. Furthermore, multiple models of HMGN1 overexpression are used in cell culture and in vivo, and both loss of function and gain of function experiments were performed in multiple models to validate the key observations. Although I did not see the initial submission, the manuscript appears to be considerably improved by these additions. Overall, I find the work to be of broad interest, the data of high quality, and the manuscript to be well written.

Thank you for stepping in to evaluate our manuscript revision and for your helpful comments below. We agree with all and have made edits to the final manuscript accordingly, which we think have indeed improved its clarity.

Below, I have a few minor comments for the authors' consideration.

1. In several panels, heatmaps are plotted as row minima and maxima (Fig. 2b, c, d, f; Fig. 3d). This representation obscures the actual differences among samples, which may be high for some genes and low for others. The authors should consider plotting these as $\log_2(\text{fold change})$ values relative to a common reference such as WT_0h.

This was a very helpful suggestion. We have replotted the listed heatmaps as \log_2 fold change relative to WT_0h. The divergence between WT and HMGN1-overexpressing cells at baseline is clearer now, as are the changes over time. New Fig 2c and 2d are pasted below as examples.

2. In the Fig. 2e metagene plots, are the data aggregated over all ATAC-seq peaks of either genotype, all differential peaks (both increased and decreased in OE cells), or only peaks increased in OE cells? If only differential peaks or increased peaks are plotted, I suggest showing two metagene plots: one for all peaks (which may show minimal overall changes in accessibility) and one for differential peaks.

The original ATAC-seq metagene panel plotted all differential peaks, both UP and DOWN, in OE cells compared to wild-type. We have now also included a metagene plot of all peaks, as suggested by the Reviewer, which shows a minimal overall change in accessibility in OE cells, and one for differential peaks, which shows the weighting toward increased differential peaks in OE cells. New Figure 2e is pasted below for reference.

3. Fig. 2d suggests that even in OE cells, Hox genes are silenced after 96 hours of differentiation. Therefore, the differentiation block in these cells is likely due to the combined effects of numerous gene expression changes. The authors should comment on this more explicitly.

This is a great point and we did not intend to assign all of the HMGN1 effect to Hox genes. Although, interestingly, when we replotted the Fig 2d heatmap as relative to WT_0h, as suggested by the Reviewer, it is clearer that the degree of Hox silencing during differentiation is not as much in the OE cells as in WT (pasted below). Nonetheless, we agree that other genes are almost certainly involved in the HMGN1 phenotype, and so we also edited the text to better acknowledge that the effects are likely due to combined effects of numerous gene expression changes.

4. In Fig. 3b or the figure legend, the authors should specify what ChIP-seq data is being depicted. From context, it appears that genes with different H3K27ac levels are examined here, but this should be stated directly since both H3K27ac and H3K27me3 profiling were performed.

Yes, this should have been labeled H3K27ac ChIP-seq. We edited the figure and legend to make this clear.

5. P14: "...(LTC-IC) assays, an in vitro measure of HSPC activity that also correlates with poorer risk in human AML41..." Does "poorer risk" mean "higher risk" and/or "poorer survival"? This term seems unnecessarily confusing.

We agree this was poorly worded. We edited the text to make it clearer.